# Revisiting Orbital Minimization Method for Neural Operator Decomposition

**J. Jon Ryu**, **Samuel Zhou**, **Gregory W. Wornell**
Department of EECS, MIT, Cambridge, MA 02139, United States
{jongha,samtzhou,gww}@mit.edu

## Abstract

Spectral decomposition of linear operators plays a central role in many areas of machine learning and scientific computing. Recent work has explored training neural networks to approximate eigenfunctions of such operators, enabling scalable approaches to representation learning, dynamical systems, and partial differential equations (PDEs). In this paper, we revisit a classical optimization framework from the computational physics literature known as the *orbital minimization method* (OMM), originally proposed in the 1990s for solving eigenvalue problems in computational chemistry. We provide a simple linear-algebraic proof of the consistency of the OMM objective, and reveal connections between this method and several ideas that have appeared independently across different domains. Our primary goal is to justify its broader applicability in modern learning pipelines. We adapt this framework to train neural networks to decompose positive semidefinite operators, and demonstrate its practical advantages across a range of benchmark tasks. Our results highlight how revisiting classical numerical methods through the lens of modern theory and computation can provide not only a principled approach for deploying neural networks in numerical simulation, but also effective and scalable tools for machine learning.

## 1 Introduction

Spectral decomposition of linear operators is a foundational tool in applied mathematics, with far-reaching implications across machine learning (ML) and scientific computing. Eigenfunctions of differential, integral, and graph-based operators capture essential geometric and dynamical structures, playing a central role in problems ranging from representation learning to solving partial differential equations (PDEs). As such, the design of efficient and scalable methods for approximating these eigenfunctions has become a core pursuit in modern computational science.

Recent advances have leveraged neural networks to approximate spectral components of operators, offering promising avenues for scaling classical techniques to high-dimensional or irregular domains in scientific simulation. Notable examples include recent breakthroughs in quantum chemistry based on neural-network ansatzes [41, 9, 42]. Beyond quantum chemistry, neural approaches have also proven effective in diverse domains such as spectral embeddings [15, 43], Koopman operator theory for analyzing dynamical systems [32, 23, 20], and neural solvers for PDEs [26]. For an overview of the broader literature, we refer the reader to [43, Appendix B]. However, many existing methods rely on surrogate losses or architectural constraints that lack a clear variational foundation, often leading to brittle optimization or limited extensibility.

In this work, we revisit a classical optimization framework from computational quantum chemistry known as the *orbital minimization method* (OMM). Originally developed in the 1990s for finding the ground state in electronic-structure theory [38, 34, 33, 37, 1], OMM offers a direct and elegant approach to approximating the top eigenspace of a positive-definite operator, without requiring

explicit orthonormalization. Despite its heuristic, domain-specific origins, we derive OMM rigorously from a simple variational principle valid for general positive-semidefinite matrices and operators.

We leverage this reinterpretation to construct a modern neural variant of OMM, suitable for decomposing a wide class of linear operators. Our contributions are threefold:

1. We provide a new derivation of the OMM objective with a simple linear-algebraic proof for finite-dimensional cases, clarifying its key idea and extending its theoretical foundation. When specialized for streaming PCA, we identify its connection to the celebrated Sanger's rule, which is often known as the generalized Hebbian algorithm [44].

2. We adapt this framework to train neural networks that learn eigenspaces of positive-definite operators, eliminating the need for explicit orthogonalization or eigensolvers.

3. We empirically demonstrate the effectiveness of our method across a range of tasks, including learning Laplacian-based representations in reinforcement learning settings, solving PDEs, and self-supervised-learning representation of images and graphs.

We emphasize that properly positioning the OMM within the modern ML literature is of both theoretical and practical significance. Although the resulting formulation may appear natural in hindsight, recent work in ML has proposed a variety of alternative, and often considerably more complex, approaches for computing eigenvectors of matrices or eigenfunctions of operators. In this work, we draw connections between OMM and several existing lines of research, including streaming PCA [44] from theoretical statistics and computational science, as well as recent neural parameterizations used for operator learning in deep learning [43], and independent development of spectral techniques in reinforcement learning literature [51, 13]. These connections provide a unified perspective on seemingly disparate methods. Our findings underscore the value of revisiting classical numerical techniques through the lens of modern ML, offering both conceptual clarity and practical advantages for scalable spectral learning.

**Notation.** For an integer $n \geq 1$, $[n] := \{1, \ldots, n\}$. We use capital sans-serif fonts, such as $\mathsf{A}$ and $\mathsf{V}$, to denote matrices. In particular, $\mathsf{I}_k \in \mathbb{R}^{k \times k}$ denotes the identity matrix. For a square matrix $\mathsf{A}$, $\mathrm{tr}(\mathsf{A})$ denotes its trace. For a matrix $\mathsf{V} \in \mathbb{R}^{d_1 \times d_2} = [\mathbf{v}_1, \ldots, \mathbf{v}_{d_2}]$, where $\mathbf{v}_j \in \mathbb{R}^{d_1}$ for each $j \in [d_2]$, we use the subscript notation $\mathsf{V}_{1:k} := [\mathbf{v}_1, \ldots, \mathbf{v}_k]$ for $1 \leq k \leq d_2$ to denote the subset of columns.

## 2 Methods

In this section, we revisit the orbital minimization method and provide a new variational perspective for the finite-dimensional matrix case. We then present its application to an infinite-dimensional problem, i.e., an operator problem.

### 2.1 Preliminary: Matrix Eigenvalue Problem

Let $\mathsf{A} \in \mathbb{R}^{d \times d}$ be a real symmetric matrix of interest.[1] We wish to compute the top-$k$ eigenvectors of $\mathsf{A}$. By the spectral theorem, $\mathsf{A}$ admits an eigenvalue decomposition, i.e., there exists an orthogonal eigenbasis $\mathsf{W} = [\mathbf{w}_1, \ldots, \mathbf{w}_d] \in \mathbb{R}^{d \times d}$ such that we can write $\mathsf{A} = \mathsf{W}\Lambda\mathsf{W}^\mathsf{T} = \sum_{i=1}^d \lambda_i \mathbf{w}_i \mathbf{w}_i^\mathsf{T}$, where $\Lambda := \mathrm{diag}(\lambda_1, \ldots, \lambda_d)$, where $\lambda_1 \geq \lambda_2 \geq \ldots \geq \lambda_d$. Our goal is to find the top-$k$ eigensubspace, i.e., $\mathrm{span}\{\mathbf{w}_1, \ldots, \mathbf{w}_k\}$, and ideally the top-$k$ eigenvectors $\mathbf{w}_1, \ldots, \mathbf{w}_k$ and corresponding eigenvalues $\lambda_1, \ldots, \lambda_k$. We use a shorthand notation $a \wedge b := \min\{a, b\}$ for $a, b \in \mathbb{R}$.

For a symmetric matrix $\mathsf{A} \in \mathbb{R}^{d \times d}$, the most standard variational characterization of the top-$k$ eigensubspace is the *multicolumn Rayleigh quotient maximization* formulation, which is

$$\max_{\mathsf{V} \in \mathbb{R}^{d \times k}: \mathsf{V}^\mathsf{T}\mathsf{V} = \mathsf{I}_k} \mathrm{tr}(\mathsf{V}^\mathsf{T}\mathsf{A}\mathsf{V}). \tag{1}$$

Note that it is a natural extension of the maximum Rayleigh quotient characterization of the largest eigenvalue. While we can derive its unconstrained version as

$$\max_{\mathsf{V} \in \mathbb{R}^{d \times k}} \mathrm{tr}((\mathsf{V}^\mathsf{T}\mathsf{V})^{-1}\mathsf{V}^\mathsf{T}\mathsf{A}\mathsf{V}), \tag{2}$$

the inversion $(\mathsf{V}^\mathsf{T}\mathsf{V})^{-1}$ renders the optimization less practical. For example, if $\mathsf{V}$ is not full rank (or nearly degenerate), computing $(\mathsf{V}^\mathsf{T}\mathsf{V})^{-1}$ may be infeasible or numerically unstable.

---

[1] We focus on real matrices for simplicity, but the technique can be naturally adapted for Hermitian matrices.

## 2.2 Orbital Minimization Method: Matrix Version

We now restrict our focus on a positive-semidefinite (PSD) matrix $A \in \mathbb{R}^{d \times d}$. The orbital minimization method (OMM) [38, 37, 34, 33, 30] aims to solve the following maximization problem:

$$\min_{V \in \mathbb{R}^{d \times k}} \mathcal{L}_{\text{omm}}(V), \quad \text{where } \mathcal{L}_{\text{omm}}(V) := -\text{tr}((2I_k - V^\mathsf{T}V)V^\mathsf{T}AV) \tag{3}$$

While it is an unconstrained objective, it is widely known that its minimizer $V^\star$ satisfies $V^\star(V^\star)^\mathsf{T} = W_{1:k}W_{1:k}^\mathsf{T}$, i.e., $V$ corresponds to the top-$k$ eigenvectors up to rotation. Interestingly, the OMM objective does not have any spurious local minima, that is, any local minima is a global optima; see [30, Theorem 2]. This approach was originally proposed to develop linear-scaling algorithms for electronic-structure calculations, i.e., algorithms whose complexity scales linearly with the number of electrons in the structure, by avoiding explicit enforcement of the orthogonality constraint $V^\mathsf{T}V = I$.

### 2.2.1 Existing Derivations

There exist two standard ways to motivate the OMM objective [10]. The first is a Lagrange-multiplier approach [38]. The Lagrangian of the constrained optimization formulation in Eq. (1) is $\mathcal{L}(V, \Xi) := \text{tr}(V^\mathsf{T}AV) - \text{tr}(\Xi(V^\mathsf{T}V - I_k))$, where $\Xi \in \mathbb{R}^{k \times k}$ denotes the Lagrange multiplier. From the KKT conditions, the optimal Lagrange multiplier is $\Xi^* = V^\mathsf{T}AV$, and the Lagrangian $\mathcal{L}(V, \Xi^*)$ yields the OMM objective.

Another argument is based on the finite-order approximation of the Neumann series expansion of the multicolumn Rayliegh quotient $\text{tr}((V^\mathsf{T}V)^{-1}V^\mathsf{T}AV)$ [10]. Assuming that the spectrum of $V^\mathsf{T}V$ is bounded by 1, we have the Neumann series expansion of the inverse matrix $(V^\mathsf{T}V)^{-1} = (I_k - (I_k - V^\mathsf{T}V))^{-1} = \sum_{k=0}^{\infty}(I_k - V^\mathsf{T}V)^k$. If we consider the first-order approximation, then $(V^\mathsf{T}V)^{-1} \approx I_k + (I_k - V^\mathsf{T}V) = 2I_k - V^\mathsf{T}V$, which yields the OMM objective:

$$\text{tr}((V^\mathsf{T}V)^{-1}V^\mathsf{T}AV) \approx \text{tr}((2I_k - V^\mathsf{T}V)V^\mathsf{T}AV) = -\mathcal{L}_{\text{omm}}(V).$$

Interestingly, in their original paper, Mauri et al. [34] showed that a higher-order extension following the same idea is possible. For any integer $p \geq 1$, define $Q_p := \sum_{i=0}^{2p-1}(I - V^\mathsf{T}V)^i$, which can be understood as the order $(2p-1)$-th order approximation of $(V^\mathsf{T}V)^{-1}$. Then, the global minimizers of

$$\tilde{\mathcal{L}}_{\text{omm}}^{(p)}(V) := -\text{tr}(Q_pV^\mathsf{T}AV) = -\text{tr}\left(\sum_{i=0}^{2p-1}(I - V^\mathsf{T}V)^iV^\mathsf{T}AV\right), \tag{4}$$

which we will refer to by the OMM-$p$ objective, also span the same top-$k$ eigensubspace. The original argument to prove this fact in [34] is domain-specific and rather obscure. Below, we provide a purely linear-algebraic and simple proof for the consistency of the OMM-$p$ objective.

### 2.2.2 New Derivation with Simple Proof

To introduce a more intuitive way to derive the general OMM objective without resorting to the Neumann series expansion, we start from noting that we can rewrite the OMM objective function as

$$\mathcal{L}_{\text{omm}}(V) = -\text{tr}((2I_k - V^\mathsf{T}V)V^\mathsf{T}AV) = \text{tr}((I_d - VV^\mathsf{T})^2A) - \text{tr}(A).$$

Hence, Eq. (3) is equivalent to minimizing $\text{tr}((I_d - VV^\mathsf{T})^2A)$. With this reformulation, we can extend the optimization problem in Eq. (3) to a higher order as

$$\mathcal{L}_{\text{omm}}^{(p)}(V) := \text{tr}((I_d - VV^\mathsf{T})^{2p}A) - \text{tr}(A) = \text{tr}\left(\left\{\sum_{j=1}^{2p}(-1)^j\binom{2p}{j}(V^\mathsf{T}V)^{j-1}\right\}V^\mathsf{T}AV\right) \tag{5}$$

for an integer $p \geq 1$. While it looks different from the expression in Eq. (4), we can show that they are exactly same, i.e., $\mathcal{L}_{\text{omm}}^{(p)}(V) = \tilde{\mathcal{L}}_{\text{omm}}^{(p)}(V)$; we include its proof in Appendix A for completeness. As stated below, the global minima of the OMM-$p$ objective characterizes the desired top-$k$ eigensubspace for any $p \geq 1$. Here we outline the main idea, leaving the rigorous proof to Appendix A.

**Theorem 1** (Consistency of the OMM-$p$ objective). *Let $r \leq d$ denote the rank of a PSD matrix $A$ with EVD $A = \sum_{i=1}^{r} \lambda_i \mathbf{w}_i \mathbf{w}_i^\mathsf{T}$. Then,*

$$\min_{V \in \mathbb{R}^{d \times k}} \mathcal{L}_{\text{omm}}^{(p)}(V) = -\sum_{i=1}^{k \wedge r} \lambda_i.$$

*In particular, its minimizer $V^\star$ satisfies $V_{1:k \wedge r}^\star(V_{1:k \wedge r}^\star)^\mathsf{T} = W_{1:k \wedge r}W_{1:k \wedge r}^\mathsf{T}$.*

*Proof sketch.* The proof crucially relies on Eq. (5), which shows that the objective depends on $V$ only through the PSD matrix $VV^\intercal \in \mathbb{R}^{d \times d}$ of rank at most $k$. Hence, we can reparameterize the optimization problem with a matrix $P \in \mathbb{R}^{d \times k}$ such that $P^\intercal P$ and a diagonal matrix $\Sigma \in \mathbb{R}^{k \times k}$ with nonnegative entries, by considering the reduced SVD of $VV^\intercal = P\Sigma P^\intercal$. We can then show that $\mathrm{tr}((I_d - VV^\intercal)^{2p}A) \geq \sum_{i=k+1}^{d} \lambda_i$, and the equality is achieved if and only if $VV^\intercal = WW^\intercal$. $\qquad\square$

### 2.2.3 Discussions

**Can We Apply OMM to Non-PSD Matrices?** Consider a rank-1 case with $p = 1$, i.e., $\mathcal{L}_{\mathrm{omm}}^{(1)}(\mathbf{v}) = \mathrm{tr}((I - \mathbf{v}\mathbf{v}^\intercal)^2 A) - \mathrm{tr}(A)$. Suppose that $A$ has a negative eigenvalue $\lambda_j$ with a normalized eigenvector $\mathbf{w}_j$. If we restrict $\mathbf{v} = c\mathbf{w}_j$ for some $c \in \mathbb{R}$, it is easy to show that we can simplify the objective as $\mathcal{L}_{\mathrm{omm}}^{(1)}(c\mathbf{w}_j) = (1 - c^2)^2 \lambda_j - \lambda_j$. Since $\lambda_j < 0$, the objective will diverge to negative infinity when $c^2 \to \infty$. This explains why the OMM cannot be applied to non-PSD symmetric matrices. If a lower bound on the smallest eigenvalue is known a priori, i.e.,, $\lambda_d \geq -\kappa$, we can *shift* the spectrum by considering the positive semidefinite matrix $A + \kappa I_d$, to which the OMM can then be applied.

**Equivalence of Regularization and Spectrum Shift.** The OMM implicitly enforces orthogonality without requiring any explicit regularization. Nevertheless, one might be tempted to introduce an additional regularization term that promotes the orthonormality of $V$, i.e., $V^\intercal V = I_k$. If we adopt a squared-Frobenius-norm penalty $\kappa \|V^\intercal V - I_k\|_F^2$ to the OMM-1 objective, it is straightforward to verify that this is equivalent to applying the OMM-1 to the spectrum-shifted matrix $A + \kappa I_d$. This reveals a peculiar property of the OMM-1: unlike standard regularization techniques that modify the optimal solution to enhance stability, the OMM-1 with this form of regularization preserves the global optima.

**Connection to PCA.** For a PSD matrix $A \in \mathbb{R}^{d \times d}$, consider a random vector $\mathbf{x} \in \mathbb{R}^d$ with mean zero and covariance $\mathbb{E}_{p(\mathbf{x})}[\mathbf{x}\mathbf{x}^\intercal] = A$. Then, we can express $\mathrm{tr}((I_d - VV^\intercal)^2 A) = \mathbb{E}_{p(\mathbf{x})}[\|\mathbf{x} - VV^\intercal \mathbf{x}\|^2]$. This is comparable to the standard characterization of PCA:

$$\min_{V \in \mathbb{R}^{d \times k}} \mathbb{E}_p[\|\mathbf{x} - V(V^\intercal V)^{-1} V^\intercal \mathbf{x}\|^2]. \tag{6}$$

If $V \in \mathbb{R}^{d \times k}$ were orthogonal, then $VV^\intercal \mathbf{x}$ is the best projection of $\mathbf{x}$ onto the column subspace of $V$, and the OMM objective could be interpreted as the corresponding approximation error. It is worth emphasizing that, even without explicitly enforcing the orthogonality constraint on $V \in \mathbb{R}^{d \times k}$, the global optima of the OMM remain unchanged, in sharp contrast to the common understanding that the whitening operation $(V^\intercal V)^{-1}$ in Eq. (6) is required to characterize the top-$k$ eigensubspace.

### 2.2.4 Nesting for Learning Ordered Eigenvectors

To learn the ordered eigenvectors, we can apply the idea of *nesting* in [43], which was originally proposed for their low-rank approximation (LoRA) objective; see Section 2.3.2 for the comparison of the LoRA and OMM. The same logic applies to the OMM. There exist two versions, joint nesting and sequential nesting, which we explain below.

The *joint nesting* aims to minimize a single objective $\mathcal{L}_{\mathrm{omm}}^{(p)}(V; \boldsymbol{\alpha}) := \sum_{i=1}^{k} \alpha_i \mathcal{L}_{\mathrm{omm}}^{(p)}(V_{1:i})$ for a choice of positive weights $\alpha_1, \ldots, \alpha_k > 0$, aiming to solve the OMM problem for $V_{1:i}$ for each $i \in [k]$. When $p = 1$, we can use the same masking technique of [43], without needing to compute the objective going over a for loop over $i \in [k]$. Similar to [43, Theorem 3.3], we have:

**Theorem 2.** *Let $V^\star \in \mathbb{R}^{d \times k}$ be a global minimizer of $\mathcal{L}_{\mathrm{omm}}^{(p)}(V; \boldsymbol{\alpha})$. For any positive weights $\boldsymbol{\alpha} \in \mathbb{R}_{>0}^k$, if the top-$(k + 1)$ eigenvalues are all distinct, $V^\star = W$.*

The proof is straightforward: if the objective can have the minimum possible value if and only if $\mathcal{L}_{\mathrm{omm}}^{(p)}(V_{1:i})$ is minimized by satisfying $V_{1:i}V_{1:i}^\intercal = W_{1:i}W_{1:i}^\intercal$ for each $i \in [k]$, which is equivalent to $\mathbf{v}_i = \mathbf{w}_i$ for each $i$.[2] Per the suggestion in [43], we use the uniform weighting $\boldsymbol{\alpha} = (\frac{1}{k}, \ldots, \frac{1}{k})$ throughout.

---

[2]Here, the strict spectral gap is assumed for simplicity. With degenerate eigenvalues, the columns corresponding to the degenerate part of an optimal $V^\star$ will be an orthonormal eigenbasis of the degenerate eigensubspace.

The idea of *sequential nesting* is to iteratively update $\mathbf{v}_i$, using its gradient of $\mathcal{L}_{\text{omm}}^{(1)}(\mathsf{V}_{1:i})$, i.e.,

$$\nabla_{\mathbf{v}_i}\mathcal{L}_{\text{omm}}^{(1)}(\mathsf{V}_{1:i}) = -2\Big((\mathsf{I}_d - \mathsf{V}_{1:i}\mathsf{V}_{1:i}^{\mathsf{T}})\mathsf{A}\mathbf{v}_i + \mathsf{A}(\mathsf{I}_d - \mathsf{V}_{1:i}\mathsf{V}_{1:i}^{\mathsf{T}})\mathbf{v}_i\Big), \tag{7}$$

as if $\mathsf{V}_{1:i-1} = [\mathbf{v}_1, \ldots, \mathbf{v}_{i-1}]$ already converged to the top-$(i-1)$ eigensubspace, for each $i \in [k]$. The intuition for convergence is based on an inductive argument.

We refer to the nested variants of OMM collectively as *NestedOMM*, and denote them by $\text{OMM}_{\text{jnt}}$ and $\text{OMM}_{\text{seq}}$ for brevity. We note that the $\text{OMM}_{\text{seq}}$ for finite-dimensional matrices was proposed as the *triangularized orthogonalization-free method (OFM)* in the numerical linear algebra literature [10].

### 2.2.5  Connection to Sanger's Algorithm for Streaming PCA

There is a rather separate line of literature on streaming PCA with long history, which aims to solve the essentially same eigenproblem for a finite-dimensional case. The goal of the streaming PCA problem is to find the leading eigenvectors of the covariance matrix $\mathsf{A} \leftarrow \mathbb{E}_{p(\mathbf{x})}[\mathbf{x}\mathbf{x}^{\mathsf{T}}]$ of a zero-mean random variable $\mathbf{x} \sim p(\mathbf{x})$. In the streaming case, we are restricted to receive a minibatch of independent and identically distributed (i.i.d.) samples $\{\mathbf{x}_1^{(t)}, \ldots, \mathbf{x}_B^{(t)}\}$ at each time $t$, and can thus only compute the empirical estimate $\mathsf{A}_t := \frac{1}{B}\sum_{b=1}^{B}\mathbf{x}_b^{(t)}(\mathbf{x}_b^{(t)})^{\mathsf{T}}$. In this setup, Sanger [44] proposed the following iterative update procedure, often called *Sanger's rule* or *generalized Hebbian algorithm* [11, 2]: at each time step $t$, for each $i = 1, \ldots, k$,

$$\mathbf{v}_i^{(t+1)} \leftarrow \mathbf{v}_i^{(t)} + \eta_t\big(\mathsf{I} - \mathsf{V}_{1:i}^{(t)}(\mathsf{V}_{1:i}^{(t)})^{\mathsf{T}}\big)\mathsf{A}_t\mathbf{v}_i^{(t)}.$$

We remark that the OMM gradient in Eq. (7) is derived from the well-defined objective and can be viewed as the *symmetrized* version of the Sanger update term, while the update term itself cannot be viewed as a gradient of any function, as its Jacobian is not symmetric; see [11, Proposition K.2]. With this connection between the OMM and Sanger's rule, as a practical variant, we can also treat the Sanger update $(\mathsf{I} - \mathsf{V}_{1:i}^{(t)}(\mathsf{V}_{1:i}^{(t)})^{\mathsf{T}})\mathsf{A}_t\mathbf{v}_i^{(t)}$ as a pseudo-gradient and plug-in to an off-the shelf gradient-based optimization algorithm, such as Adam [22]. Interestingly, in our experiment with operator learning, the Sanger variant works well in one PDE example where the OMM exhibits numerical instability. We attribute the instability of the OMM gradient to statistical noise, which may make the operator appear to have eigenfunctions with negative eigenvalues, thus triggering the divergent behavior discussed in Section 2.2.3. See Section 3.2 for a resolution and further discussion.

We note that this connection is rather surprising, given the extensive body of work over the past few decades on developing efficient streaming spectral decomposition algorithms under orthogonality constraints; see, e.g., [36, 25, 44, 31, 48, 11, 12, 51]. It is remarkable that a classical yet simple idea originating from computational chemistry provides such an elegant solution to this long-studied spectral decomposition problem—yet, to the best of our knowledge, it has remained largely unnoticed in the modern machine learning literature.

### 2.3  Orbital Minimization Method: Operator Version

In many applications, we are often tasked to compute the leading eigenfunctions of a *positive-semidefinite operator* $\mathcal{T}: L_\rho^2(\mathcal{X}) \to L_\rho^2(\mathcal{X})$. Here, $L_\rho^2(\mathcal{X}) \triangleq \{f: \mathcal{X} \to \mathbb{R}\mid \int_{\mathcal{X}} f(\mathbf{x})^2\rho(d\mathbf{x}) < \infty\}$ is a Hilbert space equipped with the inner product $\langle f_0, f_1\rangle_\rho \triangleq \int_{\mathcal{X}} f_0(\mathbf{x})f_1(\mathbf{x})\rho(d\mathbf{x})$. By the spectral theorem, if $\mathcal{T}$ is compact, then there exists a sequence of orthonormal eigenfunctions $\{f_i\}_{i\geq 1}$ with non-increasing eigenvalues $\lambda_1 \geq \lambda_2 \geq \cdots \geq 0$. When $\mathcal{X} = \{1, \ldots, d\}$ and $\rho$ is the counting measure, the space $L_\rho^2(\mathcal{X})$ is naturally identified with $\mathbb{R}^d$, and $\mathcal{T}$ reduces to a $d \times d$ PSD matrix, recovering the standard finite-dimensional eigenvalue problem in the previous section.

We can easily extend the OMM objective in Eq. (5) to this infinite-dimensional case, expressing the objective compactly using only $k \times k$ matrices. Note that the objective in Eq. (5) is a function of the *overlap matrix* $\mathsf{V}^{\mathsf{T}}\mathsf{V}$ and *projected matrix* $\mathsf{V}^{\mathsf{T}}\mathsf{A}\mathsf{V}$. In the infinite-dimensional function case, the overlap matrix and the projected matrix become the second moment matrices $\mathsf{M}_\rho[\mathbf{f}] := \int \mathbf{f}(\mathbf{x})\mathbf{f}(\mathbf{x})^{\mathsf{T}}\rho(d\mathbf{x})$ and $\mathsf{M}_\rho[\mathbf{f}, \mathcal{T}\mathbf{f}] := \int \mathbf{f}(\mathbf{x})(\mathcal{T}\mathbf{f})(\mathbf{x})^{\mathsf{T}}\rho(d\mathbf{x})$, respectively. Hence, by plugging-in this expression to Eq. (5), we can easily derive the corresponding objective for the infinite-dimensional case:

$$\mathcal{L}_{\text{omm}}^{(p)}(\mathbf{f}) := \text{tr}\left(\left\{\sum_{j=1}^{2p}(-1)^j\binom{2p}{j}(\mathsf{M}_\rho[\mathbf{f}])^{j-1}\right\}\mathsf{M}_\rho[\mathbf{f}, \mathcal{T}\mathbf{f}]\right). \tag{8}$$

In particular, if $p = 1$, it boils down to $\mathcal{L}_{\text{omm}}^{(1)}(\mathbf{f}) = -2\text{tr}(\mathsf{M}_\rho[\mathbf{f}, \mathcal{T}\mathbf{f}]) + \text{tr}(\mathsf{M}_\rho[\mathbf{f}]\mathsf{M}_\rho[\mathbf{f}, \mathcal{T}\mathbf{f}])$. If the point spectrum of $\mathcal{T}$ is isolated and separated by a spectral gap from its essential spectrum, we can also argue that the global minimizer of the OMM objective corresponds to the top-$k$ eigensubspace.

### 2.3.1 Nesting for Operator OMM

As explained in the finite-dimensional case, the idea of sequential nesting is to update the $i$-th eigenfunction $f_i$ by the gradient $\partial_{f_i}\mathcal{L}_{\text{omm}}^{(p)}(\mathbf{f}_{1:i})$ for each $i = 1, \ldots, k$. We can implement these gradients succinctly via autograd for the special case of $p = 1$. In order to do so, we define a *partially stop-gradient* second moment matrix

$$\mathsf{M}_\rho^{\text{seq}}[\mathbf{f}, \mathbf{g}] := \begin{bmatrix} \langle f_1, g_1 \rangle_\rho & \langle \text{sg}[f_1], g_2 \rangle_\rho & \langle \text{sg}[f_1], g_3 \rangle_\rho & \cdots & \langle \text{sg}[f_1], g_k \rangle_\rho \\ \langle f_2, \text{sg}[g_1] \rangle_\rho & \langle f_2, g_2 \rangle_\rho & \langle \text{sg}[f_2], g_3 \rangle_\rho & \cdots & \langle \text{sg}[f_2], g_k \rangle_\rho \\ \langle f_3, \text{sg}[g_1] \rangle_\rho & \langle f_3, \text{sg}[g_2] \rangle_\rho & \langle f_3, g_3 \rangle_\rho & \cdots & \langle \text{sg}[f_3], g_k \rangle_\rho \\ \vdots & \vdots & \ddots & \vdots \\ \langle f_k, \text{sg}[g_1] \rangle_\rho & \langle f_k, \text{sg}[g_2] \rangle_\rho & \langle f_k, \text{sg}[g_3] \rangle_\rho & \cdots & \langle f_k, g_k \rangle_\rho \end{bmatrix}.$$

Then, if we define the surrogate objective

$$\mathcal{L}_{\text{omm}}^{\text{seq}}(\mathbf{f}) := -2\text{tr}(\mathsf{M}_\rho^{\text{seq}}[\mathbf{f}, \mathcal{T}\mathbf{f}]) + \text{tr}(\mathsf{M}_\rho^{\text{seq}}[\mathbf{f}]\mathsf{M}_\rho^{\text{seq}}[\mathbf{f}, \mathcal{T}\mathbf{f}]), \tag{9}$$

then $\partial_{f_i}\mathcal{L}_{\text{omm}}^{\text{seq}}(\mathbf{f}_{1:k}) = \partial_{f_i}\mathcal{L}_{\text{omm}}^{(1)}(\mathbf{f}_{1:i})$ for each $i \in [k]$. In other words, the gradient for OMM with sequential nesting can be efficiently implemented by this *single* surrogate objective function in Eq. (9). Note that the surrogate objective is equivalent to the OMM objective $\mathcal{L}_{\text{omm}}^{(1)}(\mathbf{f}_{1:k})$ in its nominal value, but the gradients are different due to the stop-gradients. The Sanger variant can be implemented similarly with autograd; see Appendix B.

This can be implemented efficiently with almost no additional computational overhead compared to computing $\mathsf{M}_\rho[\mathbf{f}, \mathbf{g}]$ without nesting. Note that the simple implementation is made possible thanks to the fact that $\text{tr}(\mathsf{M}_\rho[\mathbf{f}]\mathsf{M}_\rho[\mathbf{f}, \mathcal{T}\mathbf{f}]) = \langle \mathsf{M}_\rho[\mathbf{f}], \mathsf{M}_\rho[\mathbf{f}, \mathcal{T}\mathbf{f}] \rangle$. For $p > 1$, however, we can no longer apply a similar, partial stop-gradient trick to the higher-order interaction terms $\text{tr}(\mathsf{M}_\rho[\mathbf{f}]^{j-1}\mathsf{M}_\rho[\mathbf{f}, \mathcal{T}\mathbf{f}])$ for $j \geq 3$, as they cannot be simply written as a matrix inner product with stop-gradient terms. Thus, it is harder to utilize the advantages of high-order OMM in the nested case. We thus only experiment with the higher-order extension in the self-supervised representation learning setting (Section 3.3), where the ordered structure is not often necessary.

Similar to the argument in [43], we can also implement OMM$_{\text{jnt}}$ for operator with $p = 1$ as follows:

$$\mathcal{L}_{\text{omm}}^{\text{jnt}}(\mathbf{f}; \boldsymbol{\alpha}) := \sum_{i=1}^{k} \alpha_i \mathcal{L}_{\text{omm}}^{(1)}(\mathbf{f}_{1:i}) = \text{tr}\left(\mathsf{P} \odot \left(-2\mathsf{M}_\rho[\mathbf{f}, \mathcal{T}\mathbf{f}] + \mathsf{M}_\rho[\mathbf{f}]\mathsf{M}_\rho[\mathbf{f}, \mathcal{T}\mathbf{f}]\right)\right).$$

Here, we define the matrix mask $\mathsf{P} \in \mathbb{R}^{k \times k}$ as $\mathsf{P}_{ij} = \mathsf{m}_{\max\{i,j\}}$ with $\mathsf{m}_i := \sum_{j=i}^{k} \alpha_j$ and $\odot$ denotes the entrywise multiplication. While Ryu et al. [43] suggest to use joint nesting for jointly parameterized eigenfunctions, we have empirically found that the convergence with sequential nesting is comparable to or sometimes better than joint nesting even with joint parameterization in general.

### 2.3.2 Comparison to Low-Rank Approximation

As alluded to earlier, a closely related approach is the low-rank-approximation (LoRA) approach based on the Eckart–Young–Mirsky theorem [8, 35] (or Schmidt theorem [45]). This approach has been widely studied in both numerical linear algebra [29, 53, 10] and machine learning [52, 15, 43, 55, 24, 20]. For a self-adjoint operator $\mathcal{T}$, LoRA seeks a rank-$k$ approximation minimizing the error in the Hilbert–Schmidt norm $\|\mathcal{T} - \sum_{i=1}^{k} f_i \otimes f_i\|_{\text{HS}}^2 - \|\mathcal{T}\|_{\text{HS}}^2$, which simplifies to:

$$\mathcal{L}_{\text{lora}}(\mathbf{f}) := -2\,\text{tr}\big(\mathsf{M}_\rho[\mathbf{f}, \mathcal{T}\mathbf{f}]\big) + \text{tr}\big(\mathsf{M}_\rho[\mathbf{f}]\mathsf{M}_\rho[\mathbf{f}]\big),$$

$$\mathcal{L}_{\text{omm}}^{(1)}(\mathbf{f}) := -2\,\text{tr}\big(\mathsf{M}_\rho[\mathbf{f}, \mathcal{T}\mathbf{f}]\big) + \text{tr}\big(\mathsf{M}_\rho[\mathbf{f}]\mathsf{M}_\rho[\mathbf{f}, \mathcal{T}\mathbf{f}]\big),$$

where we repeat $\mathcal{L}_{\text{omm}}^{(1)}(\mathbf{f})$ for direct comparison. While both objectives are unconstrained and admit unbiased gradient estimates with minibatch samples, they differ in their geometric and spectral interpretations. Let $\mathcal{T}_k := \sum_{i=1}^{k} \lambda_i \phi_i \otimes \phi_i$ denote the rank-$k$ truncation of $\mathcal{T}$. If $\mathcal{T}$ is PSD, the optimizer

$\mathbf{f}_{\mathrm{omm}}^{\star}$ of the OMM objective satisfies $\sum_{i=1}^{k} f_{\mathrm{omm},i}^{\star} \otimes f_{\mathrm{omm},i}^{\star} = \sum_{i=1}^{k} \phi_i \otimes \phi_i$, whereas the LoRA optimizer satisfies $\sum_{i=1}^{k} f_{\mathrm{lora},i}^{\star} \otimes f_{\mathrm{lora},i}^{\star} = \mathcal{T}_{k \wedge r_+}$, where $r_+$ is the number of positive eigenvalues of $\mathcal{T}$, for any self-adjoint, compact $\mathcal{T}$ [43, Theorem C.5]. In words, OMM approximates the projection operator onto the top-$k$ eigensubspace, whereas LoRA reconstructs the best rank-$k$ approximation of $\mathcal{T}$ itself. Moreover, the LoRA principle naturally extends to the singular value decomposition of general compact operators, leading to the objective $\mathcal{L}_{\mathrm{lora}}^{\mathrm{svd}}(\mathbf{f}, \mathbf{g}) = -2\mathsf{tr}(\mathsf{M}_{\rho_0}[\mathbf{f}, \mathcal{T}\mathbf{g}]) + \mathsf{tr}(\mathsf{M}_{\rho_0}[\mathbf{f}]\mathsf{M}_{\rho_1}[\mathbf{g}])$, as studied in [43]. In contrast, extending the OMM to handle general operators is nontrivial.

Interestingly, as shown in Appendix C.3.2, OMM can outperform LoRA in certain settings, particularly when the target matrix is sparse, such as the graph Laplacians of real-world networks, highlighting its robustness in structured problems. We finally remark that sequentially nested formulations of both LoRA and OMM were previously proposed and analyzed for finite-dimensional matrices [10], yet their operator-level and learning-based generalizations remain largely unexplored.

## 3 Experiments

We evaluate the OMM and its variants (including the Sanger variant) across three experimental setups. In the first two, the eigenfunctions are well defined with clear operational meaning, and estimating them in the order of eigenvalues is desirable. These results underscore the strong potential of NestedOMM for both modern machine learning applications and ML-based scientific simulations. In the third setup, we explore the applicability of OMM in self-supervised representation learning, where preserving the underlying spectral structure is often secondary to optimizing downstream task performance. Our PyTorch implementation is available at `https://github.com/jongharyu/neural-svd`. We defer the experiment details and additional numerical results to Appendix C.

### 3.1 Laplacian Representation Learning for Reinforcement Learning

We consider the representation learning problem in reinforcement learning (RL), which is known as the successor representation [5, 31] or more recently the Laplacian representation [13]. The high-level idea is that, given a transition kernel from an RL environment, we aim to find a good representation of a state in the given state space such that it reflects the intrinsic geometry of the environment.

More formally, we consider a discrete state space $\mathcal{S} = [N]$ for simplicity, but the treatment below can be easily extended to a continuous state space. Suppose that we are given an environment $p(y|x, a)$, which denotes the transition probability from $s_t = x$ to $s_{t+1} = y$ given an action $a_t = a$, from an RL problem. For a given policy $\pi(a|x)$, we consider the policy-dependent transition kernel defined as $p_\pi(y|x) := \mathbb{E}_{\pi(a|x)}[p(y|x, a)]$. In matrix notation, we denote this by $\mathsf{P}_\pi \in \mathbb{R}^{\mathcal{S} \times \mathcal{S}}$, where $(\mathsf{P}_\pi)_{xy} := p_\pi(y|x)$. In the literature [54, 51, 13], a *Laplacian* is defined as any matrix $\mathsf{L} = \mathsf{I} - f(\mathsf{P}_\pi)$, where $f$ is some function that maps $\mathsf{P}_\pi$ to a symmetric matrix, such as $f(\mathsf{P}) := \frac{1}{2}(\mathsf{P} + \mathsf{P}^\mathsf{T})$. In the experiment below, $\mathsf{P}_\pi$ is constructed to be symmetric. Like in the graph Laplacian, it is shown in the literature that the top-eigenvectors of $\mathsf{L}$ (or equivalently bottom-eigenvectors of $\mathsf{I} - \mathsf{L} = f(\mathsf{P}_\pi)$) well capture the geometry of the RL problem [31, 54].

In this experiment, we compare NestedOMM ($\mathrm{OMM}_{\mathrm{seq}}$ and $\mathrm{OMM}_{\mathrm{jnt}}$) with a recently proposed algorithm called the *augmented Lagrangian Laplacian objective* (ALLO) method [13]. While the OMM involves no tunable hyperparameters, the ALLO requires selecting a barrier coefficient $b$ and a barrier growth rate $\alpha_{\mathrm{barrier}}$ via grid search on a subset of grid environments, after which the chosen values are used for full-scale training with more epochs and transition samples.

We consider the same suite of experiments from [13], which consists of several grid environments as visualized in Appendix C.1. For each environment, we generated $10^6$ transition samples from a uniform random policy and uniform initial state distribution, used the $(x, y)$ coordinates as inputs to a neural network, and aimed to learn the top $k = 11$ eigenfunctions. We trained for $80,000$ epochs using Adam [22] with learning rate $10^{-3}$. Additionally, we reproduce the ALLO training with identical number of transition samples, epochs, and optimizer, with the suggested initial barrier coefficient $b = 2.0$ and $\alpha_{\mathrm{barrier}} = 0.01$ reported in [13]. We trained with NestedOMM for four random runs after shifting the spectrum by $\mathsf{I}$, and ALLO with 10 runs. The performance is measured by the average of cosine similarities of each mode, where degenerate spaces are handled by an oracle knowing the degeneracy.

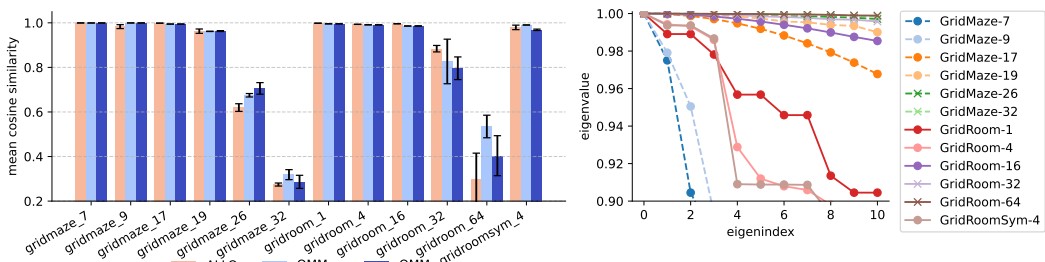

*Figure 1:* Summary of experimental results of the Laplacian representation learning. On the right panel, we plot the top-11 eigenvalues of the Laplacians. It shows that the hard instances (i.e., GridMaze-{26,32} and GridRoom{32,64}) have very small spectral gaps (marked with x). Error bar indicates one standard deviation.

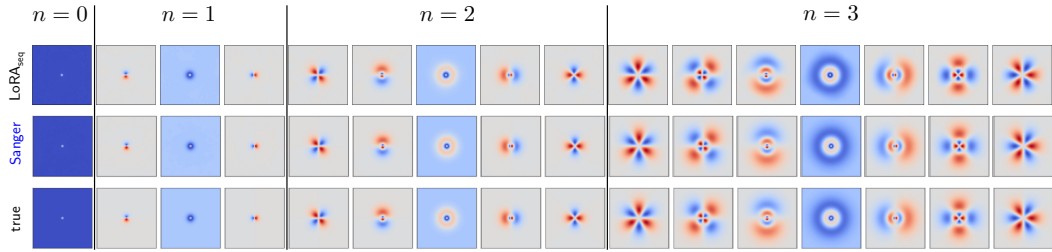

*Figure 2:* Visualization of learned eigenfunctions for the 2D hydrogen atom. OMM$_{seq}$ performs as well as LoRA$_{seq}$, which is also known as NeuralSVD$_{seq}$ [43].

We summarize the results in Figure 1. Across different environments, both OMM$_{seq}$ and OMM$_{jnt}$ perform comparably to the complex optimization dynamics of ALLO, without any hyperparameter tuning. To understand the failure cases, we plot the top-11 eigenvalues of the Laplacians on the right panel. It shows that these hard instances have very small spectral gaps, suggesting that a small cosine similarity does not necessarily indicate a failure of learning in such cases.

## 3.2 Solving Schrödinger Equations

Following [43], we now consider applying OMM to solve some simple instances of time-independent Schrödinger's equation [14]. The equation is $(\mathcal{H}\psi)(x) = \lambda\psi(x)$, where $\mathcal{H} = -\nabla^2 + V(\mathbf{x})$ is the Hamiltonian operator of a system, where $V(\mathbf{x})$ is a potential function. Since low-energy eigenfunctions correspond to the most stable states, we are primarily interested in retrieving the smallest eigenvalues of $\mathcal{H}$. Accordingly, in our convention, we aim to retrieve the eigenvalues and eigenfunctions of the operator $\mathcal{T} = -\mathcal{H}$ from the top.

### 3.2.1 Bounded-Spectrum Case

The OMM is directly applicable to certain problems, where the Hamiltonian $\mathcal{H}$ has eigenenergies bounded above, such that $\mathcal{T} = -\mathcal{H}$ (or its shifted version) is PSD. To assess the potential in such settings, we consider the simplest example of this kind: the two-dimensional hydrogen atom, where the potential function is given by $V(\mathbf{x}) = -\frac{1}{|\mathbf{x}|_2}$ for $\mathbf{x} \in \mathbb{R}^2$ [40, 43]. In this case, the true eigenvalues of $\mathcal{T}$ are expressed as $\lambda_{n,\ell} = (2n+1)^{-2}$ with two quantum numbers $n \geq 0$ and $-n \leq \ell \leq n$.

We compare the performance of OMM$_{seq}$ to LoRA$_{seq}$. We first note that we empirically found that the OMM applied to $-\mathcal{H}$ leads to unstable training. We conjecture that this instability is due to the fast-decaying spectrum and statistical noise during minibatch optimization. That is, in the beginning of optimization, if some small eigenvalue modes are captured by the model, and if the moment matrices formed by minibatch samples yield negative eigenvalues due to statistical noise, the model may diverge, as we discussed in Section 2.2. We found that this misbehavior of OMM in this example can be rectified by decomposing the shifted operator $\mathcal{T} + \kappa I$ for some $\kappa > 0$, such that the smallest eigenvalue is bounded away from zero. Instead of reporting results with OMM$_{seq}$ with identity shift, here we report the result with the Sanger variant. Rather surprisingly, we find that the

Sanger variant does not exhibit optimization instability and performs on par with LoRA$_{\text{seq}}$, which is shown to outperform existing baselines such as SpIN [40] and NeuralEF [7]. As shown in Figure 2, the learned eigenfunctions from both Sanger and LoRA$_{\text{seq}}$ are well-aligned with the ground truth, showing the competitiveness of the OMM compared to the state-of-the-art approach. Figure 4 in Appendix C provides the quantitative evaluation.

### 3.2.2 Unbounded-Spectrum Case

As alluded to earlier, the OMM is not directly applicable when the energy spectrum of $\mathcal{H}$ is unbounded above, as in the cases of the harmonic oscillator or the infinite well [14, 43].[3] To render the OMM applicable in such settings, we introduce an *inverse-operator trick*, analogous to that used in the *inverse iteration* [49]. Let $\mathcal{L}$ be a self-adjoint operator on a Hilbert space $\mathcal{H}$ with a purely discrete, positive spectrum $0 < \lambda_1 \leq \lambda_2 \leq \cdots$ with $\lambda_i \to \infty$, and corresponding orthonormal eigenfunctions $\{\phi_i\}_{i \geq 1}$. This assumption holds, for example, for Schrödinger operators on bounded domains with Dirichlet boundary conditions, or more generally, for confining potentials $V(\mathbf{x}) \to \infty$ as $\|\mathbf{x}\| \to \infty$, such as the harmonic oscillator on $\mathbb{R}^d$. In this case, $\mathcal{L}$ is invertible, and its inverse $\mathcal{L}^{-1}$ is a bounded, self-adjoint, and compact operator with eigenvalues $\lambda_i^{-1} \downarrow 0$ and the same eigenfunctions. Consequently, the top-$k$ eigenfunctions of $\mathcal{L}^{-1}$ coincide with the bottom-$k$ eigenfunctions of $\mathcal{L}$. Thus, applying the OMM to $\mathcal{L}^{-1}$ enables recovery of the lowest-$k$ modes of $\mathcal{L}$.[4]

A practical difficulty is that $\mathcal{L}^{-1}$ is rarely available in closed form; for differential operators, it corresponds to an integration operator. To avoid explicit inversion, we can parameterize $\mathbf{f} := \mathcal{L}\mathbf{g}$ with $\mathbf{g}$ represented by a neural network. Substituting this into the OMM-1 objective yields

$$\mathcal{L}_{\text{omm}}^{\text{inv}}(\mathbf{g}; \mathcal{L}) \triangleq -2\,\text{tr}\big(\mathsf{M}_\rho[\mathbf{f}, \mathcal{L}^{-1}\mathbf{f}]\big) + \text{tr}\big(\mathsf{M}_\rho[\mathbf{f}]\,\mathsf{M}_\rho[\mathbf{f}, \mathcal{L}^{-1}\mathbf{f}]\big)$$
$$= -2\,\text{tr}\big(\mathsf{M}_\rho[\mathcal{L}\mathbf{g}, \mathbf{g}]\big) + \text{tr}\big(\mathsf{M}_\rho[\mathcal{L}\mathbf{g}]\,\mathsf{M}_\rho[\mathcal{L}\mathbf{g}, \mathbf{g}]\big),$$

which preserves the same computational complexity as the original OMM. Since the $i$-th eigenfunction of $\mathcal{L}^{-1}$ is $\phi_i$, $\mathcal{L}g_i$ corresponds to $\phi_i$, and ideally $g_i = \lambda_i^{-1}\phi_i$ holds for each $i \in [k]$.

In Appendix C.2, we empirically validate this approach on the 2D infinite-well and harmonic-oscillator systems, both of which admit analytical solutions. We find that while the OMM with the inverse-operator trick accurately recovers the top eigenfunctions, it can introduce numerical instabilities and requires careful hyperparameter tuning, likely due to the nature of the parametrization.

### 3.3 Self-Supervised Contrastive Representation Learning

We now apply the OMM for contrastive representation learning. Here, we primarily consider self-supervised image representation learning using OMM, comparing to SimCLR [3], deferring its application to graph data to Appendix C.3.2. In this setup, we are given an unlabeled dataset of images $x \sim p(x)$, and our goal is to learn a network $\mathbf{f}_\theta(\cdot)$ parametrized by $\theta$ such that it yields a useful representation for downstream tasks such as classification. Following SimCLR, we consider a random augmentation $p(t|x)$ and draw two random transformations $(t_1, t_2) \sim p(t_1|x)p(t_2|x)$ to produce two views of a data entry $x$. By repeating this procedure for each image, we obtain the full dataset, which can be viewed as samples from the joint distribution $p(t_1, t_2) := \mathbb{E}_{p(x)}[p(t_1|x)p(t_2|x)]$.

The key object is then the canonical dependence kernel (CDK) $k(t_1, t_2) := \frac{p(t_1, t_2)}{p(t_1)p(t_2)}$, where $p(t) := \mathbb{E}_{p(x)}[p(t|x)]$ [43]. Most contrastive learning approaches aim to learn $\mathbf{f}_\theta(\cdot)$ such that it factorizes the log CDK (or pointwise mutual information) $\mathbf{f}_\theta(t_1)^\intercal\mathbf{f}_\theta(t_2) \approx \log k(t_1, t_2)$ [27, 28, 50]. Few exceptions include [15, 55, 43], which instead aim to factorize the CDK itself $\mathbf{f}_\theta(t_1)^\intercal\mathbf{f}_\theta(t_2) \approx k(t_1, t_2)$, such that the optimal $\mathbf{f}_\theta(t)$ can be interpreted as the low-rank approximation of the CDK. Application of the OMM to CDK provides a rather unique way to learn representations, as it does not directly aim to approximate a function of the CDK, but rather trying to find *normalized eigenfunctions* purely from a linear-algebraic point of view. Note that we can apply the OMM framework in this

---

[3]For instance, consider the Schrödinger equation under zero Dirichlet boundary conditions with the square infinite-well potential, $V(\mathbf{x}) = 0$ for $\mathbf{x} \in [-1, 1]^2$ and $V(\mathbf{x}) = \infty$ otherwise. Its eigenvalues are given by $\lambda_{n_x, n_y} \propto n_x^2 + n_y^2$, parameterized by quantum numbers $n_x, n_y \in \mathbb{N}$, which diverge as $n_x$ or $n_y \to \infty$.

[4]When the spectrum includes a continuous component whose infimum is zero (as in unconfined systems), $\mathcal{L}^{-1}$ may not be bounded. In such cases, one may instead use $(\mathcal{L} + \varepsilon I)^{-1}$ with some $\varepsilon > 0$.

*Table 1:* Top-1 and top-5 classification accuracies (%) for the SimCLR and OMM variants on CIFAR-100.

| Method | With projector | | DirectCLR (top-64 dim.) | |
|---|---|---|---|---|
| | Top-1 | Top-5 | Top-1 | Top-5 |
| OMM ($p = 1$) | 60.02 | 87.13 | 59.83 | 86.65 |
| OMM$_{jnt}$ ($p = 1$) | 61.30 | 85.22 | 59.92 | 85.13 |
| OMM$_{seq}$ ($p = 1$) | 59.91 | 87.02 | 52.96 | 81.22 |
| OMM ($p = 2$) | 63.92 | 89.08 | 61.27 | 87.07 |
| OMM ($p = 1$) + OMM ($p = 2$) | 64.77 | 89.18 | 63.99 | 88.88 |
| SimCLR [3] | 66.50 | 89.28 | N/A | N/A |

*symmetric* self-supervised setup, as the CDK is symmetric and PSD by construction. The purpose of this experiment is not to establish the state-of-the-art performance, but to assess the potential.

We followed the standard setup of [4]. We used ResNet-18 as our backbone model and adopted two different feature encoding strategies: (1) use a nonlinear projector with $2048$ hidden dimensions and $k = 256$ output dimensions as our input to the OMM; (2) in a vein similar to DirectCLR [21], take the top $k = 64$ dimensions as the feature fed to the OMM objective without a projector. In all cases, we normalized features along the feature dimension before computing any loss following the standard convention. We report the results of top-$\{1,5\}$ classification accuracy with linear probe.

The results are summarized in Table 1. We first observe that the OMM-1 yields a reasonably effective representation for classification, achieving a top-1 accuracy of approximately $60\%$. We then examined the effect of nesting on representation quality and observed no improvement, with a slight degradation in some cases. Subsequently, we evaluated the effect of the higher-order OMM with $p = 2$ as well as the mixed-order OMM combining $p \in \{1, 2\}$. Somewhat surprisingly, both variants substantially improved performance, reaching approximately $64\%$ top-1 accuracy. In Appendix D, we provide a theoretical explanation for the benefit of higher-order OMM, based on a gradient analysis. In short, the higher-order OMM gradient may provide an additional gradient to escape an immature flat local minima. A similar trend is observed with the DirectCLR variant, and notably, the performance remains comparable even without a projector. While these methods do not surpass the SimCLR baseline ($\sim 66.5\%$), the preliminary findings underscore the promise of this linear-algebraic representation learning framework and motivate further investigation.

## 4   Concluding Remarks

In this paper, we revisited a classical optimization framework from computational chemistry for computing the top-$k$ eigensubspace of a PSD matrix. Despite its appealing properties that align well with modern machine learning objectives, this approach remains underexplored in the current literature. We hope that this work stimulates further research into this linear-algebraic perspective and inspires the development of more principled learning methods. We also note that existing parametric approaches to spectral decomposition exhibit notable limitations, particularly a lack of theoretical understanding regarding their convergence properties compared to classical numerical methods. We refer an interested reader to a general discussion on the potential advantage and limitation of the parametric spectral decomposition approach over classical methods in [43].

The experimental results presented in this work are preliminary, and we believe that a more comprehensive investigation across diverse application domains could yield deeper insights and more impactful findings. In particular, extending the current framework to research-level quantum excited-state computation problems, as recently explored via alternative variational principles [42, 9], would be of significant theoretical and practical interest. Furthermore, advancing the linear-algebraic perspective on modern representation learning may open up a pathway toward more structured, interpretable, and theoretically grounded representations.

## Acknowledgments

This work was supported in part by the MIT-IBM Watson AI Lab under Agreement No. W1771646.

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

# Appendix

## A  Deferred Proofs

### A.1  Equivalence of $\mathcal{L}_{\text{omm}}^{(p)}(\mathsf{V})$ and $\tilde{\mathcal{L}}_{\text{omm}}^{(p)}(\mathsf{V})$

To establish the equivalence between the original proposal $\mathcal{L}_{\text{omm}}^{(p)}(\mathsf{V})$ in Eq. (4) and our derivation $\tilde{\mathcal{L}}_{\text{omm}}^{(p)}(\mathsf{V})$ in Eq. (5), we need to show that

$$-\text{tr}(\mathsf{Q}_p \mathsf{V}^{\intercal}\mathsf{A}\mathsf{V}) = \text{tr}((\mathsf{I}_d - \mathsf{V}\mathsf{V}^{\intercal})^{2p}\mathsf{A}) - \text{tr}(\mathsf{A}),$$

where we recall

$$\mathsf{Q}_p := \sum_{i=0}^{2p-1} (\mathsf{I}_k - \mathsf{V}^{\intercal}\mathsf{V})^i.$$

To show this, we first note that we can write

$$\mathsf{Q}_p = \sum_{i=0}^{2p-1} (\mathsf{I}_k - \mathsf{S})^i = \mathsf{S}^{-1}(\mathsf{I}_k - (\mathsf{I}_k - \mathsf{S})^{2p}),$$

by letting $\mathsf{S} := \mathsf{V}^{\intercal}\mathsf{V}$. Hence, we have

$$
\begin{aligned}
-\text{tr}(\mathsf{Q}_p\mathsf{V}^{\intercal}\mathsf{A}\mathsf{V}) &= -\text{tr}\Big(\mathsf{S}^{-1}(\mathsf{I}_k - (\mathsf{I}_k - \mathsf{S})^{2p})\mathsf{V}^{\intercal}\mathsf{A}\mathsf{V}\Big) \\
&= -\text{tr}(\mathsf{S}^{-1}\mathsf{V}^{\intercal}\mathsf{A}\mathsf{V}) + \text{tr}\Big(\mathsf{S}^{-1}(\mathsf{I}_k - \mathsf{S})^{2p}\mathsf{V}^{\intercal}\mathsf{A}\mathsf{V}\Big) \\
&= -\text{tr}(\mathsf{S}^{-1}\mathsf{V}^{\intercal}\mathsf{A}\mathsf{V}) + \text{tr}(\mathsf{S}^{-1}\mathsf{V}^{\intercal}\mathsf{A}\mathsf{V}) + \text{tr}\Big((\mathsf{I}_d - \mathsf{V}\mathsf{V}^{\intercal})^{2p}\mathsf{A}\Big) - \text{tr}(\mathsf{A}) \\
&= \text{tr}\Big((\mathsf{I}_d - \mathsf{V}\mathsf{V}^{\intercal})^{2p}\mathsf{A}\Big) - \text{tr}(\mathsf{A}).
\end{aligned}
$$

This concludes the proof.  □

### A.2  Proof of Theorem 1

*Proof of Theorem 1.* Since $\mathsf{V}\mathsf{V}^{\intercal}$ is of rank at most $k$ and PSD, the optimization can be reparameterized based on the reduced SVD of $\mathsf{V}\mathsf{V}^{\intercal} = \mathsf{P}\Sigma\mathsf{P}^{\intercal}$, where $\mathsf{P} := [\mathbf{p}_1, \ldots, \mathbf{p}_k] \in \mathbb{R}^{d \times k}$ and

$\Sigma := \text{diag}(\mu_1, \ldots, \mu_k) \in \mathbb{R}^{k \times k}$, such that $P^{\mathsf{T}}P = I_k$ and $\mu_1 \geq \ldots \geq \mu_k \geq 0$. For a given orthogonal matrix $P \in \mathbb{R}^{d \times k}$, let $P_\perp \in \mathbb{R}^{d \times (d-k)}$ denote a matrix whose columns form an orthonormal basis of the orthogonal complement of the column subspace of P. Note $PP^{\mathsf{T}} + P_\perp P_\perp^{\mathsf{T}} = I_d$ and $P_\perp^{\mathsf{T}} P = O_{(d-k) \times k}$ by construction, so that we can write

$$(I_d - VV^{\mathsf{T}})^{2p} = (P(I_k - \Sigma)P^{\mathsf{T}} + P_\perp P_\perp^{\mathsf{T}})^{2p}$$
$$= P(I_k - \Sigma)^{2p}P^{\mathsf{T}} + P_\perp P_\perp^{\mathsf{T}}.$$

Hence, the objective function can be lower bounded as

$$\text{tr}((I_d - VV^{\mathsf{T}})^{2p}A) = \text{tr}(P(I_k - \Sigma)^{2p}P^{\mathsf{T}}A) + \text{tr}(P_\perp^{\mathsf{T}}AP_\perp)$$
$$\overset{(a)}{\geq} \text{tr}(P_\perp^{\mathsf{T}}AP_\perp)$$
$$\overset{(b)}{\geq} \sum_{\ell=k+1}^{d} \lambda_\ell.$$

Here, $(a)$ follows since $P(I_k - \Sigma)^{2p}P^{\mathsf{T}}$ and A are both PSD and the inner product of two PSD matrices is always nonnegative. Further, $(b)$ follows since $\text{tr}(P_\perp^{\mathsf{T}}AP_\perp)$ is minimized as $\sum_{\ell=k+1}^{d} \lambda_\ell$ when optimized over an orthogonal matrix $P_\perp \in \mathbb{R}^{d \times (d-k)}$. We now consider the equality condition. First, $(b)$ holds with equality if and only if $P_\perp$ consists of a bottom-$(d-k)$ eigenbasis of A, or equivalently P can be written as $P = VQ^{\mathsf{T}}$ by an orthogonal matrix $Q \in \mathbb{R}^{k \times k}$. Given that, we can write

$$\text{tr}(P(I_k - \Sigma)^{2p}P^{\mathsf{T}}A) = \text{tr}((I_k - \Sigma)^{2p}QV^{\mathsf{T}}AVQ^{\mathsf{T}})$$
$$= \text{tr}((I_k - \Sigma)^{2p}Q\Lambda_{1:k}Q^{\mathsf{T}})$$
$$= \sum_{\ell=1}^{k}(1 - \mu_\ell)^{2p}\lambda_\ell q_\ell q_\ell^{\mathsf{T}}.$$

This implies that $(a)$ holds with equality if and only if $\mu_\ell = 1$ for all $\ell \in [k \wedge r]$. $\qquad\square$

### A.3 Proof of Theorem 2

*Proof of Theorem 2.* The global minima of the jointly nested objective $\mathcal{L}_{\text{omm}}^{(p)}(V; \boldsymbol{\alpha}) := \sum_{i=1}^{k} \alpha_i \mathcal{L}_{\text{omm}}^{(p)}(V_{1:i})$ is achieved if and only if $\mathcal{L}_{\text{omm}}^{(p)}(V_{1:i})$ is minimized for each $i \in [k]$. Hence, if the $V^\star \in \mathbb{R}^{d \times k}$ achieves the global minima, then by the optimality condition from $\mathcal{L}_{\text{omm}}^{(p)}(V_{1:i}^\star)$, it must satisfy

$$\sum_{j=1}^{i} v_j^\star (v_j^\star)^{\mathsf{T}} = \sum_{j=1}^{i} w_j w_j^{\mathsf{T}}$$

for each $i \in [k]$. By telescoping, this leads to $v_i = w_i$ for each $i$. $\qquad\square$

## B   On Implementation

In this section, we present omitted details on the implementation of NestedOMM.

### B.1   Implementation of Sanger's Variant

Similar to the implementation of the sequential nesting by the following partially stop-gradient objective

$$\mathcal{L}_{\text{omm}}^{\text{seq}}(\mathbf{f}) := -2\text{tr}(M_\rho^{\text{seq}}[\mathbf{f}, \mathcal{T}\mathbf{f}]) + \text{tr}(M_\rho^{\text{seq}}[\mathbf{f}]M_\rho^{\text{seq}}[\mathbf{f}, \mathcal{T}\mathbf{f}]),$$

we can implement the Sanger variant by auto-differentiating the following objective

$$\mathcal{L}_{\text{omm}}^{\text{sanger}}(\mathbf{f}) := -2\text{tr}(M_\rho[\mathbf{f}, \mathcal{T}\mathbf{f}]) + \text{tr}(M_\rho^{\text{seq}}[\mathbf{f}]\text{sg}[M_\rho[\mathbf{f}, \mathcal{T}\mathbf{f}]]).$$

With the appropriate stop-gradient operation, its gradient implements the Sanger variant.

## B.2 Pseudocode for NestedOMM-1

Here, we include a unified PyTorch [39] implementation of various versions of the original OMM with $p = 1$ (which we call OMM-1): OMM-1 without any nesting (when nesting is None), OMM-1 with the joint nesting (when nesting is jnt), OMM-1 with the sequential nesting (when nesting is seq), and OMM-1 with the Sanger variant (when nesting is sanger).

```python
class NestedOrbitalLoss:
    def __init__(
        self,
        nesting=None,
        n_modes=None,
    ):
        assert nesting in [None, 'jnt', 'seq', 'sanger']
        self.nesting = nesting
        if self.nesting == 'jnt':
            assert n_modes is not None
            self.vec_mask, self.mat_mask = get_joint_nesting_masks(weights=np.ones(
    n_modes) / n_modes)
        else:
            self.vec_mask, self.mat_mask = None, None

    def __call__(self, f: torch.Tensor, Tf: torch.Tensor):
        # f: [b, k]
        # Tf: [b, k]
        if self.nesting == 'jnt':
            M_f = compute_second_moment(f)
            M_f_Tf = compute_second_moment(f, Tf)
            operator_term = -2 * (torch.diag(self.vec_mask.to(f.device)) * M_f_Tf).mean
    (0).sum()
            metric_term = (self.mat_mask.to(f.device) * M_f_Tf * M_f).sum()
        else:  # if self.nesting in [None, 'seq', 'sanger']:
            M_f = compute_second_moment(f, seq_nesting=self.nesting in ['seq', 'sanger'
    ])
            M_f_Tf = compute_second_moment(f, Tf, seq_nesting=self.nesting == 'seq')
            if self.nesting == 'sanger':
                M_f_Tf = M_f_Tf.detach()
            operator_term = -2 * torch.trace(M_f_Tf)
            metric_term = (M_f_Tf * M_f).sum()

        return operator_term + metric_term

def compute_second_moment(
        f: torch.Tensor,
        g: torch.Tensor = None,
        seq_nesting: bool = False,
    ) -> torch.Tensor:
    """
    compute (optionally sequentially nested) second-moment matrix
        M_ij = <f_i, g_j>
    with partial stop-gradient handling when seq_nesting is True.

    args
    ----
    f : (n, k) tensor
    g : (n, k) tensor or None
    seq_nesting : bool
    """
    if g is None:
        g = f
    n = f.shape[0]
    if not seq_nesting:
        return (f.T @ g) / n
    else:
```

```
56          # apply partial stop-gradient
57          # lower-triangular: <f_i, sg[g_j]> for i > j
58          lower = torch.tril(f.T @ g.detach(), diagonal=-1)
59          # upper-triangular: <sg[f_i], g_j> for i < j
60          upper = torch.triu(f.detach().T @ g, diagonal=+1)
61          # diagonal:          <f_i, g_i>     (no stop-grad)
62          diag  = torch.diag((f * g).sum(dim=0))
63          return (lower + diag + upper) / n
64
65
66  def get_joint_nesting_masks(weights: np.ndarray):
67      vector_mask = list(np.cumsum(list(weights)[::-1])[::-1])
68      vector_mask = torch.tensor(np.array(vector_mask)).float()
69      matrix_mask = torch.minimum(
70          vector_mask.unsqueeze(1), vector_mask.unsqueeze(1).T
71      ).float()
72      return vector_mask, matrix_mask
```

### B.3   Computational Complexity of OMM

In terms of complexity in computing a given objective, the complexity of OMM-1 is similar to LoRA [43] and ALLO [13]. For each minibatch of size $B$, the dominant factor is in computing the empirical moment matrices, which takes $O(Bk^2)$ complexity for a given minibatch of size $B$ and for $k$ eigenfunctions. When $p > 1$, the complexity becomes $O(Bk^2 + pk^3)$, as we need to recursively compute the powers of the overlap matrix in Eq. (8).

For a detailed comparison between the parametric spectral decomposition approach and a conventional numerical eigensolver, we refer to [43, Appendix A]. There, the authors report an undesirable scaling behavior exhibited by numerical eigensolvers and highlight the efficiency advantage of the LoRA-based parametric formulation. Given that the computational complexity of gradient evaluation in LoRA and OMM is equivalent, the same argument applies here.

## C   Details on Experimental Setups and Additional Results

In this section, we describe the detail on the experimental setups. All codebases to reproduce the experiments will be made public upon acceptance of the paper. All experiments were conducted on a single GPU, either a `NVIDIA GeForce RTX 3090` (24GB) or a `NVIDIA RTX A6000` (48GB).

### C.1   Laplacian Representation Learning for Reinforcement Learning

Our implementation was built upon the codebase of Gomez et al. [13].[5]

#### C.1.1   Experimental Setup

We followed the experimental setup in [13] closely. We visualize the RL environments in Figure 3.

- **Data generation**: For each experiment, we generated $N = 10^6$ transition samples from a uniform random policy and uniform initial state distribution. The $(x, y)$ coordinates are given to a neural network as inputs.
- **Architecture**: We used a fully connected neural network with 3 hidden layers of 256 units with ReLU activations, with an output layer of $k = 11$ to learn the top-$k$ Laplacian eigenfunction representation.
- **Optimization**: We trained for 80,000 epochs using the Adam optimizer with a learning rate of $10^{-3}$. In the first $10\%$ of training steps, we also add a linear warm-up from 0 to $10^{-3}$. Additionally, we reproduce the ALLO training with identical number of transition samples, epochs, and optimizer, with the suggested initial barrier coefficient $b = 2.0$ and $\alpha_{\text{barrier}} = 0.01$ reported in [13]. We trained with NestedOMM for four random runs, and ALLO with 10 runs. Since the smallest eigenvalue of each Laplacian matrix can be negative

---

[5]Github repository: `https://github.com/tarod13/laplacian_dual_dynamics`

in this case, but being bounded below by $-1$ (by the Gershgorin circle theorem), we add the identity matrix to the Laplacian matrix to shift the spectrum.

- **Performance metric**: The performance is measured by the average of cosine similarities of each mode, where degenerate spaces are handled by an oracle knowing the degeneracy.

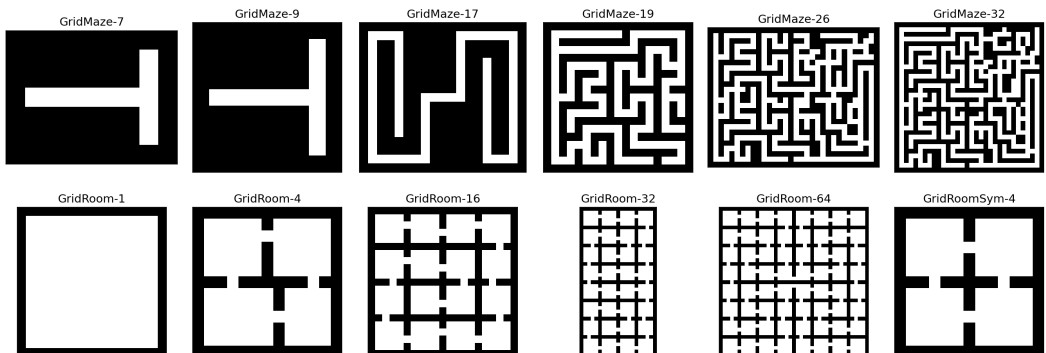

*Figure 3:* Grid environments for Laplacian representation learning.

## C.1.2 Ablation Study: Hyperparameter Sensitivity of ALLO vs OMM

We continue with the experimental setup in [13] but attempt the more difficult task of learning the top $k = 50$ rather than top $k = 11$ Laplacian eigenfunctions. Our goal is to examine differences in hyperparameter sensitivity between ALLO and OMM.

- **Revised Setup:** For both OMM and ALLO training in the ablation study, we use the following common modifications to the experimental setup: we generate $N = 5 \cdot 10^6$ transition samples from a uniform random policy and uniform initial state distribution. Our architecture keeps the 3 hidden layers of 256 units with ReLU activations, but the output layer's dimension is $k = 50$ to learn the top-50 Laplacian eigenfunctions. Additionally, we omit the GridMaze-7 and GridMaze-9 environments, as they have less than 50 eigenmodes. The optimization recipe and performance metric remain the same.

- **Initial Evaluation of ALLO Hyperparameters:** We begin by examining how well the tuned ALLO hyperparameters transfer from $k = 11$ to $k = 50$. In particular, we use the suggested initial barrier coefficient $b = 2.0$ and $\alpha_{\text{barrier}} = 0.01$. After training each environment for four runs, we obtain the results displayed in Table 2(a). It is clear that the learned eigenfunctions do not capture the top-50 modes well, and that we require additional hyperparameter tuning to effectively learn the eigenmodes with ALLO.

- **ALLO Hyperparameter Sweep:** As in [13], we sweep the initial barrier coefficient $b$ over $[0.5, 1.0, 2.0, 10.0]$, while $\alpha_{\text{barrier}}$ sweeps over $[0.001, 0.01, 0.05, 0.1, 0.2, 0.5, 1.0]$. We train over the GridRoom-1, GridRoom-16, and GridMaze-19 environments, and we also use $10^6$ transition samples instead of $2 \cdot 10^5$ for hyperparameter selection. This yields the best pair as $(b, \alpha_{\text{barrier}}) = (1.0, 0.1)$. Using the updated hyperparameters, we rerun ALLO training for $k = 50$ with 3 independent runs per environment. The results are displayed in Table 2(b). We can observe a significant jump in performance, but at the expense of a costly hyperparameter search.

- **OMM Retraining With Same Hyperparameters:** In contrast, we train OMM for $k = 50$ with the same recipe as $k = 11$, with the only difference being the increase to $5 \cdot 10^6$ transition samples. We train for 5 independent runs per environment and get the results displayed in Table 2(c). As we can see, the representational power of OMM is comparable with ALLO, but we were able to train without the costly pre-training hyperparameter sweep.

## C.2 Solving Schrödinger Equations

We followed the setup in [43] closely, implementing our code based on top of theirs.[6]

---

[6]Github repository: `https://github.com/jongharyu/neural-svd`

*Table 2:* Ablation study results with ALLO and OMM for $k = 50$.

*(a)* ALLO for $k = 50$ with optimal hyperparameters tuned for $k = 11$.

| Env | GridMaze-17 | GridMaze-19 | GridMaze-26 | GridMaze-32 | GridRoom-1 |
|---|---|---|---|---|---|
| Mean | 0.8150 | 0.7616 | 0.4361 | 0.3340 | 0.8081 |
| Std | 0.0234 | 0.0277 | 0.0361 | 0.0303 | 0.0077 |

| Env | GridRoom-4 | GridRoom-16 | GridRoom-32 | GridRoom-64 | GridRoomSym-4 |
|---|---|---|---|---|---|
| Mean | 0.6414 | 0.6692 | 0.4837 | 0.3420 | 0.6297 |
| Std | 0.0190 | 0.0153 | 0.0451 | 0.0164 | 0.0321 |

*(b)* ALLO for $k = 50$ with re-tuned hyperparameters ($b = 1.0$, $\alpha_{\text{barrier}} = 0.1$).

| Env | GridMaze-17 | GridMaze-19 | GridMaze-26 | GridMaze-32 | GridRoom-1 |
|---|---|---|---|---|---|
| Mean | 0.9798 | 0.9669 | 0.8233 | 0.7359 | 0.9088 |
| Std | 0.0022 | 0.0001 | 0.0165 | 0.0176 | 0.0328 |

| Env | GridRoom-4 | GridRoom-16 | GridRoom-32 | GridRoom-64 | GridRoomSym-4 |
|---|---|---|---|---|---|
| Mean | 0.8681 | 0.8885 | 0.7452 | 0.7417 | 0.7880 |
| Std | 0.0004 | 0.0127 | 0.0093 | 0.0473 | 0.0241 |

*(c)* OMM for $k = 50$ with same training recipe for $k = 11$.

| Env | GridMaze-17 | GridMaze-19 | GridMaze-26 | GridMaze-32 | GridRoom-1 |
|---|---|---|---|---|---|
| Mean | 0.9435 | 0.9594 | 0.8226 | 0.6536 | 0.9532 |
| Std | 0.0111 | 0.0128 | 0.0150 | 0.0370 | 0.0090 |

| Env | GridRoom-4 | GridRoom-16 | GridRoom-32 | GridRoom-64 | GridRoomSym-4 |
|---|---|---|---|---|---|
| Mean | 0.8697 | 0.9191 | 0.7433 | 0.6985 | 0.8107 |
| Std | 0.0045 | 0.0015 | 0.0219 | 0.0350 | 0.0041 |

### C.2.1 Experimental Setup for 2D Hydrogen Atom

Exactly same configurations were used for both the Sanger variant and LoRA$_{\text{seq}}$ with sequential nesting. Since the configuration is almost same as [43, Appendix E.1.1], we note the setup succinctly, and refer an interested reader to therein for the detailed setup.

- **Data generation**: We opted to use a Gaussian distribution $\mathcal{N}(0, 16\mathsf{I}_2)$ as a sampling distribution, and generated new minibatch sample of size 512.
- **Architecture**: We used 16 separate fully connected neural networks with 3 hidden layers of 128 units with the softplus activation. We also used the multi-scale Fourier features [47].
- **Optimization**: We trained the neural networks for $10^5$ iterations. To enable a direct comparison between OMM and LoRA under consistent computational settings, we used five times fewer iterations than in [43]. While longer training may improve performance, our choice facilitates a controlled evaluation of relative efficacy. We note that we used Adam optimizer [22] with learning rate $10^{-4}$ with the cosine learning rate scheduler, instead of RMSprop [17], which we found to perform worse than Adam. We multiplied the operator by 100, but did not shift it by a multiple of identity.
- **Performance metric**: Following [43], we report the relative errors in eigenvalue estimates, angle distances for each mode (also similar to the Laplacian representation learning for RL), and subspace distances within each degenerate subspace. For each metric, we followed the same procedure defined in [43]. We ran 10 different training runs for each method and report average values with standard deviations.

The numerical comparisons are summarized in Figure 4. First, we note that the OMM$_{\text{seq}}$ diverged quickly without any additional identity shift, as we alluded to earlier. The Sanger variant is comparable to or sometimes even better than LoRA$_{\text{seq}}$ (i.e., NeuralSVD$_{\text{seq}}$) in terms of different metrics. Since LoRA$_{\text{seq}}$ was the strongest method from [43], this result suggests that OMM can be also alternatively

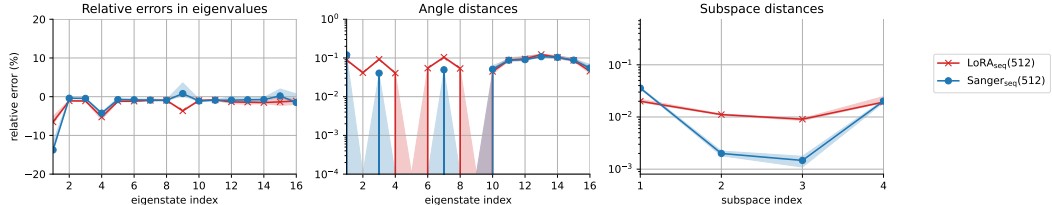

*Figure 4:* Summary of the 2D hydrogen experiment in Figure 2. The shaded region indicates $\pm$ standard deviations.

used in place of LoRA when we wish to decompose a PSD operator. As alluded to earlier, however, OMM cannot deal with unbounded differential operators (such as harmonic oscillators) inherently, while LoRA is capable of that.

### C.2.2 Results on Unbounded-Spectra Case

In this section, we present numerical results for Schrödinger equations with unbounded spectra with the inverse-operator trick.

**2D Harmonic Oscillator.** When $V(\mathbf{x}) = \|\mathbf{x}\|^2$ in the Schrödinger, it is called the quantum harmonic oscillator. This is another classical textbook example in quantum mechanics [14]. For simplicity, we consider the two-dimensional case. Similar to the 2D hydrogen atom case, we closely followed the setting in [43], except the followings.

- **Data generation**: We used a Gaussian distribution $\mathcal{N}(0, 4\mathsf{I}_2)$ as a sampling distribution with batch size 512.
- **Architecture**: We used $k = 28$ disjoint neural networks essentially same as those used for the 2D hydrogen atom. On top of that, we applied the Dirichlet boundary mask [41] over the bounded box $[-10, 10]^2$, to ensure that the parametric eigenfunctions vanish outside the box. We found that this is essential to enable successful training.
- **Optimization**: We trained for $5 \times 10^4$ iterations. We used Adam optimizer [22] with learning rate $10^{-4}$ and the cosine learning rate scheduler. We note that RMSProp did not lead to a successful training in this case.

The results are shown in Figures 5 and 6. Figure 5 visualizes the top-15 learned eigenfunctions. After applying subspace alignment via the orthogonal Procrustes procedure for fair comparison, the learned eigenfunctions closely match the ground-truth solutions. The quantitative results in Figure 6 further confirm that the corresponding eigenvalues are accurately recovered.

**2D Infinite Well.** Yet another simple example is the infinite-well problem, where $V(\mathbf{x}) = 0$ for $\mathbf{x} \in \Omega$ and $V(\mathbf{x}) = \infty$ otherwise for some domain $\Omega$. In this case, the stationary Schrödinger equation reduces to the Laplace equation on $\Omega$ with zero Dirichlet boundary conditions. We consider the two-dimensional problem with $\Omega = [-L, L]^2$ where $L = 5$. The eigenfunctions can be explicitly written as

$$\psi_{n_x, n_y}(x, y) = \frac{1}{L} \sin\left(\frac{n_x \pi (x + L)}{2L}\right) \sin\left(\frac{n_y \pi (y + L)}{2L}\right), \tag{10}$$

with eigenvalues

$$\lambda_{n_x, n_y} = \frac{\pi^2}{4L^2} (n_x^2 + n_y^2). \tag{11}$$

Here, $(n_x, n_y) \in \mathbb{N} \times \mathbb{N}$ are the quantum numbers.

- **Data generation**: We used a uniform distribution for sampling with batch size 512.
- **Architecture**: We used $k = 15$ disjoint neural networks without random Fourier features, but with the Dirichlet boundary mask.
- **Optimization**: We trained for $5 \times 10^3$ iterations. We used Adam optimizer [22] with learning rate $10^{-3}$ without any learning rate scheduler. RMSProp was also not effective

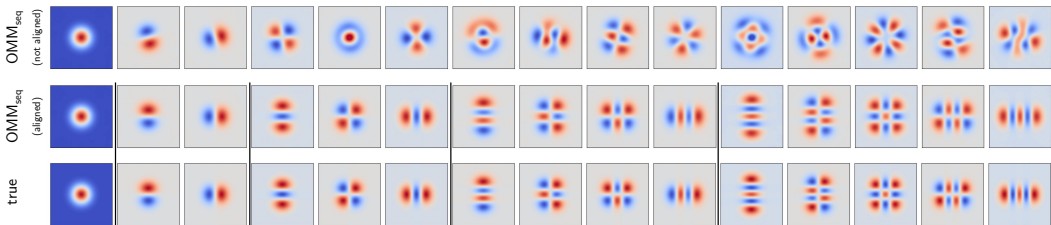

*Figure 5:* Visualization of the first 15 of learned eigenfunctions with OMM$_{seq}$ on the 2D harmonic oscillator. The first row shows the raw learned parametric eigenfunctions. The second row presents the eigenfunctions aligned to the ground-truth degenerate subspaces via the orthogonal Procrustes procedure as instructed in [43]. The third row shows the ground-truth eigenfunctions.

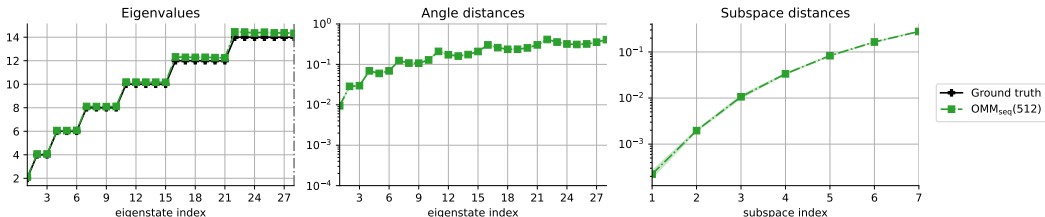

*Figure 6:* Summary of the 2D harmonic oscillator experiment in Figure 6. The shaded region indicates $\pm$ standard deviations.

in this case. Also, we found that the autograd implementation is very crucial with the Laplacian operator in the Hamiltonian, while the finite-difference-based approximation of the Laplacian, which is proven to work for other experiments and in [43], leads to slow convergence. We conjecture that this behavior is due to the operator-dependent parameterization in the operator-inverse trick, which could amplify the approximation gap of the operator, if there is any.

The results are shown in Figures 7 and 8. While the training process can be sensitive to certain hyper-parameter configurations, the learned eigenfunctions exhibit high quality once properly optimized. Such sensitivity is a common issue in deep learning practice, though it may make the OMM less appealing in comparison, as the LoRA approach is considerably more straightforward to apply.

### C.3 Self-Supervised Contrastive Representation Learning

In this section, we describe the experimental setup for the image experiment in the main text, as well as an additional graph experiment.

#### C.3.1 Representation Learning for Images

For this experiment, we used the `solo-learn` codebase of da Costa et al. [4].[7]

- **Data generation**: We used the default data augmentations for CIFAR-100 in the codebase. The exact configurations to reproduce the results will be shared upon acceptance.

- **Architecture**: We used ResNet-18 [16] as our backbone model and adopted two different feature encoding strategies: (1) we used a nonlinear projector of shape `Linear(feature_dim,2048)-BatchNorm1D-ReLU-Linear(2048,2048)-BatchNorm1D-ReLU-Linear(2048,256)`; (2) similar to DirectCLR [21], we removed the projector and simply train the top $k = 64$ dimensions of the ResNet-18 feature as the top-$k$ eigenfunctions using the OMM objective. In both cases, each feature vector is normalized by its $\ell_2$-norm following the standard convention.

---

[7]Github repository: `https://github.com/vturrisi/solo-learn`

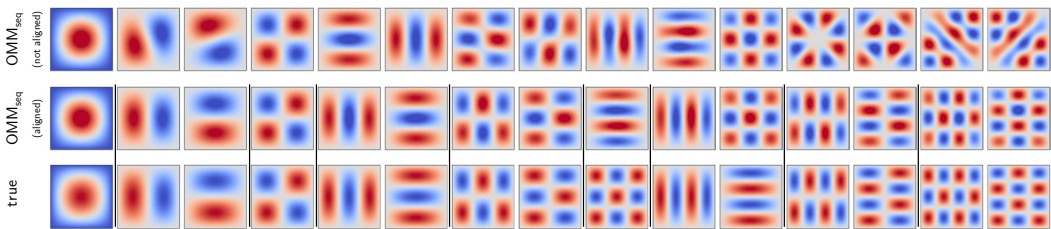

*Figure 7:* Visualization of the first 15 of learned eigenfunctions with OMM$_{seq}$ on the 2D harmonic oscillator. The first row shows the raw learned parametric eigenfunctions. The second row presents the eigenfunctions aligned to the ground-truth degenerate subspaces via the orthogonal Procrustes procedure as instructed in [43]. The third row shows the ground-truth eigenfunctions.

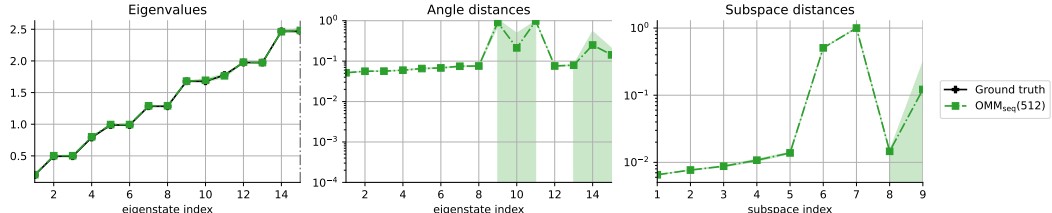

*Figure 8:* Summary of the 2D harmonic oscillator experiment in Figure 7. The shaded region indicates $\pm$ standard deviations.

- **Optimization**: We used the default optimization configuration of the codebase.[8] We use the LARS optimizer [56] with weight decay set to 0, initial learning rate of 0.3 governed by a cosine decay schedule, batch size 256, and 1000 epochs.

- **Evaluation**: We evaluate the representation based on the linear probe accuracy on the test split, trained by SGD with learning rate 0.1, batch size 256, and 100 epochs.

### C.3.2 Representation Learning with Graph Data

We can apply the orbital minimization principle to graph data. Suppose that we are given an adjacency matrix $A \in \mathbb{R}^{N \times N}$, where $A_{ij}$ encodes the connectivity between nodes $i$ and $j$. In the spectral graph theory [46], it is well known that the lowest eigenvectors of (symmetrically normalized) graph Laplacian $L_{sym} := I - D^{-1/2}AD^{-1/2}$ encodes important properties of the underlying graph, and it is also well known that the spectrum is bounded within $[0, 2]$.

When the graph is large and when each node $i \in [N]$ is associated with a feature vector $\mathbf{x}_i$, computing the eigenvectors numerically might be cumbersome and extrapolation to new points is nontrivial. Hence, in this case, it is natural to learn eigenfunctions $\mathbf{f}(\mathbf{x})$ of the graph Laplacian as a function of the feature vector $\mathbf{x}$. In this section, we show the applicability of OMM in this scenario, and assess the quality of the learned eigenfunctions of the graph Laplacian in the node classification task.

We closely followed the experimental setup in the *Neural Eigenmaps* paper [6], implementing our code based on the codebase of Deng et al. [6].[9] Neural Eigenmaps was proposed as a method to find eigenfunctions of a PSD operator similar to OMM, but it is a regularization-based approach and thus does not have the sharp global optimality that OMM enjoys. Moreover, practitioners also need to tune the regularization parameter $\alpha$ in the framework, while OMM is hyperparameter-free.

- **Data**: We used the ogbn-products dataset [18], where the feature vector has 100 dimensions and the classification task has 47 classes. It is a large-scale node property prediction benchmark, and the accompanied graph consists of 2,449,029 nodes and 61,859,140 edges. The density of this graph is $2.06 \times 10^{-5}$, suggesting that the underlying graph is extremely sparse.

---

[8]We refer the reader to the configuration for SimCLR pretraining: https://github.com/vturrisi/solo-learn/blob/main/scripts/pretrain/cifar/simclr.yaml.

[9]Github repository: https://github.com/thudzj/NEigenmaps

- **Architecture**: We parameterize the eigenfunctions by an 11-layer MLP encoder with a width of 2048 and residual connections [16], followed by a projector of the same architecture as in the image experiment. We used 8192-4096 hidden units for OMM and LoRA, and 8192-8192 for Neural Eigenmaps. We note that, with OMM, we did not require the feature normalization by $\ell_2$-norm. In contrast, Neural Eigenmaps quickly diverged without the $\ell_2$-norm normalization and thus used the normalization so that the feature has $\ell_2$-norm 10. Even worse, we also trained a model with LoRA [43] (i.e., NeuralSVD without nesting), but the training dynamics were unstable and diverged regardless of $\ell_2$-normalization.

- **Optimization**: Training was conducted over 20 epochs on the full set of nodes using the LARS optimizer [56] with a batch size of 16384, weight decay set to 0, and an initial learning rate of 0.3 with a cosine learning rate scheduler. We used the default hyperparameter $\alpha = 0.3$ for Neural Eigenmaps as suggested in the paper [6], which was selected based on linear probe accuracy on the validation set.

- **Evaluation**: Similar to the image experiment, we evaluate the representation based on the linear probe accuracy on the test split, trained by SGD with learning rate 0.01 and weight decay $10^{-3}$, batch size 256, and 100 epochs. We consider two evaluation strategies. The first is to train a linear classifier directly on the original training labels. The second follows the *Correct & Smooth* (C&S) method of Huang et al. [19], which enhances node classification by first correcting the training labels using a graph-based error estimation and then smoothing the corrected labels via feature propagation. This procedure produces a refined supervision signal for training, often leading to improved downstream performance. We used the default configuration in the Neural Eigenmaps codebase for C&S.

**Results.** We summarize our result in Table 3.

- First, unlike the standard image contrastive representation learning setting and Neural Eigenmaps, we found that OMM was capable of training the final embedding trained to fit the eigenfunctions to become highly performant on the classification task. That is, remarkably, the linear probe performance from the embedding is better than the intermediate feature, which is the output of the MLP. We note that the drastic performance drop in Neural Eigenmaps from $\sim 74\%$ (representation) to $\sim 50\%$ (embedding) is a typical behavior. This implies that while Neural Eigenmaps might provide sufficient signal for the intermediate feature to capture relevant information about each node, the *embedding* might not be truly trained to fit the underlying eigenfunctions and thus provides worse discriminative power. On the other hand, the good classification performance of embedding (even better than representation) of OMM suggests that the OMM objective may behave better than competitors in the context of capturing true eigenfunctions.

- Second, we observe that the C&S postprocessing boosts the classification accuracy for all cases to be relatively close. Nonetheless, even after the application of C&S, we find that the performance of the OMM embedding is clearly the best.

*Table 3:* Summary of OGBN-products experiment (%). The model was trained once for each method, but the linear probe were trained for 10 different times for each case. The ±'s indicate the standard deviations. "Representation" refers to the linear probe accuracy based on the output of the MLP backbone, and "Embedding" refers to that based on the output of the projector, which is trained to fit the underlying eigenfunctions. The LoRA objective failed to yield convergent training dynamics.

| | Finetuning | | Correct & Smooth [19] | |
|---|---|---|---|---|
| | representation | embedding | representation | embedding |
| LoRA [43] | N/A | N/A | N/A | N/A |
| Neural Eigenmaps [6] | $74.05_{\pm 1.71}$ | $50.76_{\pm 0.72}$ | $82.40_{\pm 0.91}$ | $80.90_{\pm 0.20}$ |
| OMM (ours) | $73.66_{\pm 1.88}$ | $\mathbf{74.17_{\pm 0.19}}$ | $82.06_{\pm 1.03}$ | $\mathbf{84.11_{\pm 0.12}}$ |

# D  On the Benefit of Higher-Order OMM

In the self-supervised image representation learning experiment, we observe sharp increase in the downstream task performance by using the OMM-2 objective $\mathcal{L}_{\text{omm}}^{(2)}(\mathsf{V})$, and even better by using the mixed objective $\mathcal{L}_{\text{omm}}^{(1)}(\mathsf{V}) + \mathcal{L}_{\text{omm}}^{(2)}(\mathsf{V})$. In this section, we provide a theoretical argument on the practical benefit of the higher-order OMM based on a gradient analysis.

We analyze the gradient for the finite-dimensional case for simplicity, but the same argument is readily extended to the function case. Let $\mathsf{R} := \mathsf{I}_d - \mathsf{VV}^\intercal$. Then, we can show, by chain rule, that the gradient of the OMM-$p$ objective is

$$
\begin{aligned}
\nabla_\mathsf{V}\mathcal{L}_{\text{omm}}^{(p)}(\mathsf{V}) &= \nabla_\mathsf{V}\text{tr}((\mathsf{I}_d - \mathsf{VV}^\intercal)^{2p}\mathsf{A}) \\
&= \nabla_\mathsf{R}\text{tr}(\mathsf{R}^{2p}\mathsf{A})\nabla_\mathsf{V}\mathsf{R} \\
&= -2\left(\sum_{i=0}^{2p-1}\mathsf{R}^i\mathsf{A}\mathsf{R}^{2p-1-i}\right)\mathsf{V} \\
&= -2(\mathsf{R}^{2p-1}\mathsf{A} + \mathsf{R}^{2p-2}\mathsf{A}\mathsf{R} + \ldots + \mathsf{R}\mathsf{A}\mathsf{R}^{2p-2} + \mathsf{A}\mathsf{R}^{2p-1})\mathsf{V}.
\end{aligned}
$$

For the purpose of our analysis, we restrict our attention to the case where $\mathsf{R} = \mathsf{VV}^\intercal$ is idempotent, i.e., $\mathsf{R}^2 = \mathsf{R}$. Then the gradient expression simplifies to

$$
\begin{aligned}
\nabla_\mathsf{V}\mathcal{L}_{\text{omm}}^{(p)}(\mathsf{V}) &= -2(\mathsf{RA} + \mathsf{AR} + (2p-2)\mathsf{RAR})\mathsf{V} \\
&= \nabla_\mathsf{V}\mathcal{L}_{\text{omm}}^{(1)}(\mathsf{V}) - 4(p-1)\mathsf{RARV},
\end{aligned}
$$

where we have the base $p = 1$ case of

$$
\nabla_\mathsf{V}\mathcal{L}_{\text{omm}}^{(1)}(\mathsf{V}) = -2(\mathsf{RA} + \mathsf{AR})\mathsf{V}
$$

Hence, if we consider the Frobenius norm of the gradient,

$$
\|\nabla_\mathsf{V}\mathcal{L}_{\text{omm}}^{(p)}(\mathsf{V})\|_\mathrm{F}^2 = \|\nabla_\mathsf{V}\mathcal{L}_{\text{omm}}^{(1)}(\mathsf{V})\|_\mathrm{F}^2 + \Delta,
$$

where we let

$$
\Delta := 16(p-1)^2\|\mathsf{RARV}\|_\mathrm{F}^2 - 8(p-1)\text{tr}\left(\nabla_\mathsf{V}\mathcal{L}_{\text{omm}}^{(1)}(\mathsf{V})^\intercal\mathsf{RARV}\right).
$$

This shows that when $p > 1$ is sufficiently large, the gradient norm can be made strictly larger than the norm of $\nabla_\mathsf{V}\mathcal{L}_{\text{omm}}^{(1)}(\mathsf{V})$. This can improve convergence speed when near convergence, especially when $\|\mathsf{I}_k - \mathsf{V}^\intercal\mathsf{V}\|_\mathrm{F}$ is close to 0, since the gradient norm of the original OMM gradient $\|\nabla_\mathsf{V}\mathcal{L}_{\text{omm}}^{(1)}(\mathsf{V})\|_\mathrm{F}$ becomes small proportional to $\|\mathsf{I}_k - \mathsf{V}^\intercal\mathsf{V}\|_\mathrm{F}$. In practical optimization, the flat minima may cause immature convergence, and the additional gradient signal from the OMM with $p > 1$ can help escape the flat minima.

