# OpenReview forum: "Revisiting Orbital Minimization Method for Neural Operator Decomposition"
_NeurIPS.cc/2025/Conference — NeurIPS 2025 poster_

### Official Review · Reviewer_cjx3 · 2025-06-21

**Clarity:** 4
**Significance:** 3
**Originality:** 4
**Rating:** 5
**Confidence:** 4

**Summary:**

This paper revisits the orbital minimization method (OMM), a classical optimization framework from computational chemistry originally developed in the 1990s, and adapts it for modern neural network-based spectral decomposition tasks. The authors provide a new theoretical derivation of the OMM objective using simple linear algebraic arguments, extending beyond the original domain-specific context to general positive semidefinite matrices and operators. They demonstrate the method's effectiveness across three application domains: Laplacian representation learning for reinforcement learning, solving Schrödinger equations, and self-supervised contrastive representation learning. To me atleast the key insight is that OMM offers an unconstrained optimization approach to eigenspace computation without requiring explicit orthogonalization, which I think makes it particularly appealing for neural network implementations.

**Questions:**

I have several questions that could help clarify the contribution and potentially improve my assessment:

1. Numerical Stability: The authors mention stability issues with the direct OMM approach for the Schrödinger equation but don't provide a thorough analysis. Can you characterize when these stability problems occur and provide theoretical or empirical guidance on when practitioners should expect issues?
2. Computational Complexity: How does the computational cost of OMM-based neural training compare to traditional eigensolvers or other spectral decomposition methods? The paper claims to avoid explicit orthogonalization, but what about the overall wall-clock time and memory requirements?
3. Hyperparameter Sensitivity: The experiments show competitive performance, but there's limited discussion of hyperparameter sensitivity. How robust is the method to choices like learning rate, network architecture, and the order parameter p in higher-order OMM variants?
4. Scalability: The experimental domains are relatively modest in scale. How do you expect the method to perform on very high-dimensional problems where traditional eigensolvers become prohibitive?

**Ethical Concerns:**

["NO or VERY MINOR ethics concerns only"]

**Final Justification:**

The authors' rebuttal successfully addressed my key concerns by providing clear explanations for numerical stability issues with practical solutions, adequate computational complexity context, and honest scalability assessments. While experimental results remain somewhat preliminary and the method is limited to PSD operators, these are appropriately acknowledged limitations. The solid theoretical contribution (clean OMM derivation, connections to existing methods) combined with comprehensive experimental validation across diverse domains outweighs the preliminary aspects, making this a valuable contribution that merits acceptance.

**Limitations:**

I think the authors do a reasonable job addressing limitations in their concluding remarks. They acknowledge that the experimental results in representation learning are preliminary and note the broader limitations of parametric spectral decomposition methods. However, I would have liked to see more discussion of the numerical stability issues and clearer guidance on when practitioners should consider using OMM versus alternatives. The restriction to PSD operators is appropriately highlighted, though I think more analysis of the spectrum shifting workaround would strengthen the paper.

**Quality:**

3

**Strengths And Weaknesses:**

Strengths: The theoretical contribution is quite solid - I appreciate how the authors provide a clean, intuitive derivation of the OMM objective that doesn't rely on the somewhat obscure Neumann series arguments from the original computational chemistry literature. The connection they draw between OMM and other methods like streaming PCA and Sanger's algorithm is illuminating and helps position the work within the broader ML landscape. The experimental validation is reasonably comprehensive, spanning three quite different application domains. I think the authors did a good job showing that the method can work across diverse settings, from discrete grid environments to quantum mechanics to image representation learning. The comparison with established baselines like ALLO in the RL experiments and LoRA in the PDE setting provides useful context. The writing is generally clear and the paper is well-structured. The authors do a nice job motivating why one might want to revisit classical numerical methods in the context of modern ML, and they're appropriately honest about limitations.

Weaknesses: The restriction to positive semidefinite operators is a significant limitation that constrains the method's applicability. While the authors acknowledge this and suggest spectrum shifting as a workaround, this feels somewhat ad-hoc and may affect the theoretical guarantees. I found some of the experimental results to be preliminary. In the representation learning experiments, the performance doesn't quite match SimCLR, and the authors seem to treat this as acceptable since it's not their main focus. However, this makes it harder to assess the practical value of the approach in this domain. The numerical stability issues mentioned in the Schrödinger equation experiments are concerning. The fact that they had to switch to the Sanger variant suggests that the core OMM method may have robustness problems that aren't fully characterized. The computational complexity analysis is somewhat lacking. While the authors mention that OMM avoids explicit orthogonalization, they don't provide a detailed comparison of computational costs versus traditional eigensolvers or other spectral methods.

---

> ### Author Rebuttal · Authors · 2025-07-31
>
> We appreciate the reviewer's time and effort in evaluating our paper and providing thoughtful feedback. We acknowledge that the concerns raised are valid and believe that addressing them will substantially enhance the quality of the manuscript. Below, we respond to each point individually.
>
> ---
> 1. `Numerical Stability: The authors mention stability issues with the direct OMM approach [...] Can you characterize when these stability problems occur and provide theoretical or empirical guidance on when practitioners should expect issues?`
> We appreciate the reviewer for raising this question. We agree that this warrants a better clarification, as we posited this as a fundamental flaw of OMM in the current writing. After we submitted our manuscript, we were able to identify that such instability can occur due to the statistical noise, if the underlying operator has diminishingly eigenvalues close to 0. (Recall that the Hamiltonian of the 2D hydrogen atom has such a property.) While the operator is PD, since we estimate the second moment matrices for step only with finite minibatch samples, there is a chance due to statistical noise and variance in estimation that the empirical moment matrix may falsely signal a negative eigenvalue. Preventing this failure mode can be done easily by adding a small amount of the “identity” shift to the target operator, such that the spectrum is strictly bounded away from zero. We empirically verified that this trick works effectively in practice. We will add a thorough ablation study on this matter in our revision.
>
> 2. `Computational Complexity: How does the computational cost of OMM-based neural training compare to traditional eigensolvers or other spectral decomposition methods? The paper claims to avoid explicit orthogonalization, but what about the overall wall-clock time and memory requirements?`
>     OMM has a similar computational complexity, as in computing the objective function, to LoRA and ALLO. A more critical question would be its comparison to the numerical eigensolver. In \[35\], where the authors advocated the LoRA objective for scientific computing and representation learning, they performed such an empirical analysis on computational complexity. As we demonstrated the comparable performance of OMM to LoRA in the example, the same conclusion can be deduced for OMM. While we borrow the discussion from \[35\] for now, we will add a similar yet more in-depth analysis on the comparison of OMM to numerical eigensolvers.
>
>     In \[Appendix A.3, 35\], the authors performed an empirical comparison of the computational complexity of the neural-network solution obtained by LoRA against one of the most representative, generic numerical eigensolver LOBPCG, for the 2D hydrogen atom problem.  They computed the eigenvalues of the Hamiltonian by increasing the grid size, and plot the accuracy of the eigenvalues and the wall-clock time. They showed that the accuracy of the numerical solver gets improved with increasing resolution, but the time complexity scaled cubically. Given this, they showed that LoRA+neural network solution achieved much higher accuracy with a shorter wall-clock time. While there exist some caveats in this comparison as noted therein (about, e.g., that such apple-to-apple comparison may not be fair and some structure of a problem at hand can be exploited), the scaling behavior of the naive application of the generic numerical eigensolver even for this simple problem clearly demonstrates the benefit of the parametric approach based on LoRA and OMM.
>
> 3. `Hyperparameter Sensitivity: The experiments show competitive performance, but there's limited discussion of hyperparameter sensitivity. How robust is the method to choices like learning rate, network architecture, and the order parameter p in higher-order OMM variants?`
>
>      We acknowledge that we currently do not have any sensitivity analysis in the empirical results, especially mostly borrowing neural network architectures and optimization configurations from prior works. In our revision, however, we will add some ablation study on the choices as well as on the effect of $p$ beyond representation learning, to provide better insights for researchers who wish to apply this methodology to new problems. Below, we briefly overview the architectures and optimization configuration.
>
>     - The architectures used in our experiments were a 3-layer MLP for Laplacian representation learning, ResNet-18 for image representation learning, disjoint 3-layer MLPs for solving Schrodinger’s equation, and a 11-layer MLP encoder with skip connections for the graph experiment in the Appendix. For the 3-layer MLP for Laplacian representation learning, we observed significantly stronger gradient signals at the beginning of training than the middle, but we opted to fix this with a linear warmup rather than architectural changes.
>
>     - We did not explicitly sweep learning rates and usually used the default learning rate/scheduler for similar approaches. However, as suggested by the strong initial gradients and linear warmup fix in the Laplacian representation learning experiment, using a scheduler to keep gradient update magnitudes comparable is important.
>
> 4. `Scalability: The experimental domains are relatively modest in scale. How do you expect the method to perform on very high-dimensional problems where traditional eigensolvers become prohibitive?`
>
>     This is an important question. To demonstrate the scalability of the method for more challenging problems, we will add some more controlled experiments with analytical solutions, such as N-dim infinite well and N-dim harmonic oscillator problems, as described in our response to **Reviewer YTWU**; see Item 4 therein. We expect the method to extend well to these simple examples, albeit high-dimensional, fairly straightforwardly, as long as we scale the architecture properly. A more challenging problem would be to solve the computational chemistry problem, which is to solve the Schrodinger equation for an atom with multiple electrons. For such a challenging problem with a much more complicated potential, the current methodology may not work as is, as efficient sampling from the high-dimensional becomes the main bottleneck of the problem. The recent breakthroughs in the field handle the issue by the quantum (variational) Monte Carlo (QMC), which resorts to a good architecture and sampling from the distribution induced by the neural network ansatz. We believe that one may be able to modify OMM to apply to this challenging problem, which could result in a very neat solution to the important problem. We leave this direction as an exciting future work.

---

> > ### Comment · Reviewer_cjx3 · 2025-08-04
> > **Response to authors**
> >
> > Thank you for your comprehensive and thoughtful responses. Based on these responses, I am raising my assessment of the paper. The core theoretical contribution remains solid, and your answers provide the practical insights I was seeking.

---

> > > ### Author Response · Authors · 2025-08-05
> > >
> > > Thank you for your follow-up and for considering a higher assessment of our paper. We’re glad that our responses helped clarify the practical aspects and supported the theoretical contribution. We appreciate your time and thoughtful feedback.

---

### Official Review · Reviewer_piiF · 2025-06-23

**Clarity:** 4
**Significance:** 4
**Originality:** 3
**Rating:** 5
**Confidence:** 4

**Summary:**

This paper breathes new life into the Orbital Minimization Method (OMM) -- a classical idea from quantum chemistry -- and repositions it as a powerful, modern tool for learning spectral representations with neural networks. The authors offer a clean and elegant re-derivation of OMM from first principles in linear algebra, showing that its objective naturally leads to orthogonal eigenvectors without any explicit constraints. This key property makes OMM uniquely well-suited for integration with deep learning. It also generalizes OMM to handle both finite-dimensional matrices and infinite-dimensional operators, and introduces neural OMM variants using a novel nesting technique to extract ordered eigenspaces. Across diverse domains -- from reinforcement learning (Laplacian-based representations), to quantum physics (solving Schrödinger equations), to self-supervised learning (representation learning from unlabeled data) -- the authors demonstrate that OMM is not only theoretically grounded but also practically robust. By unifying it with various existing spectral methods like streaming PCA and LoRA, the paper successfully convinces me that revisiting old ideas through a modern lens can yield surprisingly powerful solutions.

**Questions:**

Below are not suggestions to make changes, but something I, as a reader, would like to see more discussions with:

1. Given that OMM minimizes a surrogate objective without explicit orthogonality constraints, does the optimization introduce any systematic bias toward specific eigenvectors or bases (e.g., favoring more localized or smoother modes), especially in high multiplicity eigenspaces?

2. Can the higher-order OMM variants (p > 1) be interpreted through a statistical lens, such as variance regularization or higher-order moment matching?

**Ethical Concerns:**

["NO or VERY MINOR ethics concerns only"]

**Final Justification:**

The authors have addressed all my questions, I am keeping my positive rating.

**Paper Formatting Concerns:**

No concerns.

**Quality:**

4

**Strengths And Weaknesses:**

Overall, this is a high-quality, well-written paper that contributes both theoretical insight and practical methodology. I also think the work is important, mainly because it touches one of the foundations in ML -- eigenspace solving problem. In particular, it eliminates the need for orthogonality constraints (common in PCA/SVD-based methods). This work presents a method that integrates naturally with gradient-based deep learning, which enables scalable and stable eigenspace learning, even in operator or infinite-dimensional settings.

Weakness are minor -
1. although the core idea behind OMM was originally proposed in the 1990s within the context of quantum chemistry -- somewhat weakening the perceived algorithmic originality -- the authors’ new derivation (w/ simpler proof), reinterpretation, and adaptation to modern machine learning settings (esp. the nesting usage), bring genuine novelty and practical value.
2. Experiments are solid but not exhaustive; more large-scale results or ablations could have strengthened the case.

---

> ### Author Rebuttal · Authors · 2025-07-31
>
> We appreciate the reviewer’s effort in reviewing our paper and asking thought-provoking questions. Below, we address the reviewer’s questions.
>
> ---
> 1. `Given that OMM minimizes a surrogate objective without explicit orthogonality constraints, does the optimization introduce any systematic bias toward specific eigenvectors or bases (e.g., favoring more localized or smoother modes), especially in high multiplicity eigenspaces?`
>     This is an interesting question! We observe that OMM does not appear to introduce systematic bias of this nature, and it is capable of fitting to the true eigenfunctions provided the neural network architecture is sufficiently expressive. That being said, we can observe “smoother” modes as the reviewer suggested if we restrict the capacity of the neural network, which is independent of the OMM objective. More concretely, consider a physical system (like the hydrogen example) where the first eigenfunctions correspond to the most stable configurations. In this case, higher modes have (spatially) higher frequency components, and thus we need more “expressive” neural networks to fit to higher modes. We empirically observe that by limiting the capacity of the network via decreasing the width from 128 to 32, we can see that the learned eigenfunctions are smoother than the true eigenfunctions, while the orthogonality is still approximately preserved due to the implicit orthogonalization induced by the OMM objective. We believe that this type of experiment can provide further insights on the choice of neural network architecture, and thus we will include a similar ablation study in the appendix.
>
> 2. `Can the higher-order OMM variants (p > 1) be interpreted through a statistical lens, such as variance regularization or higher-order
> moment matching?`
>
>     This is again a very interesting question. To our understanding, since higher-order OMM variants are based on powers of the second moment matrix, they do not directly encode higher-order statistical moments. The variance regularization perspective, if there is such an effect, would be pleasantly surprising; we leave such theoretical exploration as a future direction.
>
>     We wish to remark, however, that the higher-order version has a statistical drawback in practical optimization. Namely, to estimate its gradient in an unbiased manner, which is essential in scalable optimization, we need to use at least $p$ independent samples to handle the $p$-th power of the moment matrix. Given a finite minibatch size, splitting a fixed-size minibatch into $p$ subsets may increase the variance of each individual moment estimate. Therefore, for a finite batch setting, there might exist a sweet spot in the choice of optimal $p$. We also leave such an investigation to future work.

---

> ### Comment · Reviewer_piiF · 2025-08-03
> **response to rebuttal**
>
> I would like to thank the authors to further spend time adding ablation study, as well as sharing the future plans for this branch of work.
>
> Still, I am extremely impressed by the simplicity, theoretical insights and connections that the authors drew in this paper which breathes new life into OMM. I would like to keep my positive rating in support of this work.

---

> > ### Author Response · Authors · 2025-08-05
> >
> > We appreciate the reviewer’s thoughtful support and engagement. We will carefully revise the manuscript to incorporate all the points addressed in the rebuttal.

---

### Official Review · Reviewer_YTWU · 2025-07-01

**Clarity:** 3
**Significance:** 4
**Originality:** 4
**Rating:** 5
**Confidence:** 4

**Summary:**

The authors extend the orbital minimization method (OMM) to the decomposition of positive semi-definite (psd) operators and show its applications to a range of benchmark tasks. The extension of OMM from matrices to operators is done elegantly, and the authors contextualize their work very well.

**Questions:**

See weaknesses above. Could the authors address this both in the response and in the text?

I'd be particularly curious in the case of the (negative) Laplacian on a 2D domain with some boundary conditions, where we know the true eigenfunctions (for [-1,1]^2, for instance). If you recovered the eigenfunctions using operator OMM, what accuracy would those have after sampling on a regular grid and comparing to the eigenvectors computed from, say, a p-th order finite difference discretization of that Laplacian?

I'm willing to significantly increase my score based on responses.

**Ethical Concerns:**

["NO or VERY MINOR ethics concerns only"]

**Final Justification:**

The authors addressed my questions and have run the additional experiments I requested, specifically about error estimates, numerical guidance, the transition from continuous to discrete, and the recovery of known eigenvalues and eigenvectors.

**Limitations:**

The authors do not have an explicit limitations statement. Basically, the work needs more intuitive experimental evidence that can allow sci comp and SciML practitioners to decide tradeoffs.

**Quality:**

4

**Strengths And Weaknesses:**

Strengths:

1. The paper is incredibly well-written, the language among the clearest I've seen, and the contexualization in the body of literature is fantastic.
2. Operator OMM will likely have many applications in scientific computing and scientific machine learning.
3. Connecting iterative PCA updates to gradient descent for OMM is also a very scientifically novel idea.

Weaknesses:
Despite the clarity, the authors skip over a few details without citations or exposition. I would have also liked to see more SciML or sci comp friendly experiments.

1. What is the partially stop-gradiented second moment matrix, as in, why is this matrix called that?
2. What is the object actually being parametrized by a neural network in the operator OMM? Is it the eigenfunction? If so, the paper needs a clear step where the operator OMM goes from continuous to discrete, with all the parametrizations and sampling details clearly written down (before we get to the results).
3. Does a discretized operator OMM on a continuous operator recover a particular discrete OMM result for some discretization of that operator? Is there an error estimate or perhaps a numerical guidance rule?
4. I'd be curious to see how this recovers eigenfunctions of the Laplacian Beltrami operator or of positive-definite reproducing kernels. These would be important examples for sci comp and SciML.
5. A more basic benchmark was definitely needed! For instance, in the quantum mechanics case, connecting to the Laplace-Beltrami operator and showing the recovery of orbitals for the hydrogren atom.

---

> ### Author Rebuttal · Authors · 2025-07-31
>
> We appreciate the reviewer’s effort in reviewing our paper and providing constructive feedback. We will incorporate the suggestions in our revision to improve the clarity and better demonstrate the significance of our work.
> Below, we provide answers and clarifications to the points in Weaknesses / Questions.
>
> ---
> 1. `What is the partially stop-gradiented second moment matrix, as in, why is this matrix called that?`
> We acknowledge that the explanation around the implementation of sequential nesting may be too terse. We will revise the paragraph to improve readability by adding the following explanation.
>
>     In a nutshell, the partially stop-gradiented second matrix is introduced to efficiently implement the gradients of sequential nesting via autograd, unlike the custom gradient implementation of sequential nesting for LoRA in \[41\]. As explained in the finite-dimensional case, the idea of sequential nesting is to update the $k$-th eigenfunction $f\_k$ by the gradient $\\partial\_{f\_k} \\mathcal{L}\_{\\mathsf{orbital}}(\\mathbf{f}\_{1:k})$ for each $k=1,\\ldots,K$.
>     To implement these gradients succinctly via autograd, we introduce the “partially stop-gradiented” moment matrix as in after line 205 and define a surrogate objective function for sequential nesting as in after line 204. Then, we can formally show that $$\\partial\_{f\_1,\\ldots,f\_K}\\mathcal{L}\_{\\mathsf{orbital}}^{\\mathsf{seq}}(\\mathbf{f}\_{1:K})=\\begin{bmatrix}   \\partial\_{f\_1} \\mathcal{L}\_{\\mathsf{orbital}}(f\_{1})\\\\  \\vdots\\\\  \\partial\_{f\_K} \\mathcal{L}\_{\\mathsf{orbital}}(\\mathbf{f}\_{1:K})  \\end{bmatrix},$$ which implies that we can simply compute this surrogate objective function to update a model using the gradients from sequential nesting. Note that the surrogate loss is equivalent to the OMM objective (with p=1) in its nominal value, but the gradients are different due to the stop gradient operations.
>     We call the matrix “partially stop-gradiented”, as we apply stop-gradient operations “partially” to the definition of the second moment matrix to implement the sequential nesting’s gradient.
>
> 2. `What is the object actually being parametrized by a neural network in the operator OMM? Is it the eigenfunction? If so, the paper needs a clear step where the operator OMM goes from continuous to discrete, with all the parametrizations and sampling details clearly written down (before we get to the results).`
>
>     For the operator case, a neural network is meant to parameterize the top-K eigensubspace (or the top-K eigenfunctions when nesting is applied). We agree that the exact parameterization of the eigenfunctions are of great importance in SciML, and we will revise the manuscript to clarify them in the main text. Currently all the detailed settings for the experiments can be found in Appendix C.
>
>     `Regarding “when the operator OMM goes from continuous to discrete”`:  We understand the motivation behind this question, and here is how we can understand the transition.
>     - In the ML setup, where we are given a finite (yet possibly large) dataset of size $N$, the “discretized” matrix of size $N\\times N$ is readily (but implicitly) defined, and there exists a full batch version of the objective function we wish to optimize. When we optimize it in practice, in the modern ML paradigm, we implement using minibatch, only looking at its submatrix at a time.
>     - In the SciML setup, there is a delicate difference. Consider solving a linear PDE over a given domain. In this case, instead of discretizing the domain by a fixed grid, we can freely sample from the domain according to a certain probability measure to optimize the population OMM objective in a minibatch sense. In this case, there is no “prior” discretization, and we update the model using freshly sampled data points from the domain for every minibatch. Since any practical optimization proceeds over a finite number of iterations, we may view the set of all sampled points encountered during training as inducing a discretized version of the operator, which the model effectively decomposes.
>
> 3. `Does a discretized operator OMM on a continuous operator recover a particular discrete OMM result for some discretization of that operator? Is there an error estimate or perhaps a numerical guidance rule?`
>
>     We believe the first part of this question is addressed in the previous item. The second part is particularly relevant in the SciML setup, where we have the flexibility to choose the sampling distribution. See Appendix D.3 of \[41\] for a discussion on the importance sampling variant of the LoRA objective, which applies directly to OMM. A high-level rule of thumb in choosing a good sampling distribution is that it is beneficial to choose a distribution that has more probability mass around where the true eigenfunctions have non-zero values. However, since this requires a prior knowledge of the eigenfunctions to some extent, this does not always lead to a practical numerical guidance. We believe that, as the reviewer implied, a principled error estimate or numerical rule to guide optimization would be highly desirable to make this approach more applicable, and we leave this as a future research direction.
>
> 4. `Eigenfunctions of the Laplacian Beltrami operator or of positive-definite reproducing kernels? More examples?`
>
>     We acknowledge that having more examples for PDEs will further strengthen the contribution of the paper. In short, we performed additional experiments on decomposing the 2D Laplacian operator over a bounded domain as well as 2D quantum harmonic oscillators with OMM, and were able to recover the underlying eigenfunctions to high accuracy. Below, we explain a bit of details and additional technicalities in this experiment, and we will revise the manuscript to highlight and strengthen this contribution.
>
>     - The reviewer particularly asked about solving a Laplacian over a 2D square, which is essentially a Schrodinger equation for square infinite-well potential if the boundary condition is of Dirichlet type. In this case, we have analytical solutions to the underlying eigenvalues, which is characterized by two quantum numbers $n\_x,n\_y\\in \\mathbb{N}$ as $\\lambda\_{n\_x,n\_y}\\propto n\_x^2+n\_y^2$. At first glance, as the eigenvalues are increasing and diverging to infinity as $n\_x,n\_y\\to\\infty$, it seems that we cannot apply the OMM as discussed in the main text, since we cannot negate the operator and shift it, as we cannot avoid negative parts.
>
>     - To address the issue in such operators, we propose to apply the “inverse operator” trick as follows. Let $\\mathcal{L}$ be a linear operator that has positive eigenvalues $0\<\\lambda\_1\\le \\lambda\_2\\le \\ldots $. Since this implies the invertibility of the operator, we can consider decomposing the inverse $\\mathcal{L}^{-1}$, which has eigenvalues $\\lambda\_1^{-1}\\ge \\lambda\_2^{-1} \\ge \\ldots \\ge 0$, and the top-$k$ eigenfunction $\\psi\_k$ corresponds to the bottom-$k$ eigenfunction of the original operator $\\mathcal{L}$. This implies that we can apply OMM to learn top-$k$ eigenfunctions of $\\mathcal{L}^{-1}$ to learn the bottom-$k$ eigenfunctions of $\\mathcal{L}$.
>
>     - A technical challenge is then that we may not know how to compute the inverse operator $\\mathcal{L}^{-1}$. For example, the inverse would correspond to an integration operator for the case of differential operators. To detour this challenge, we can set $\\mathbf{f}\gets \\mathcal{L}\\mathbf{g}$, where $\\mathbf{g}$ is the function we parameterize by neural networks. After all, this yields the following objective function:
>     $$  \\begin{aligned}  \\mathcal{L}\_{\\mathsf{orbital}}^{\\mathsf{inv}}(\\mathbf{g};\\mathcal{L})  &\\triangleq \-2\\mathrm{tr}(\\mathsf{M}\_{\\rho}\[\\mathbf{f},\\mathcal{L}^{-1}\\mathbf{f}\]) \+ \\mathrm{tr}(\\mathsf{M}\_{\\rho}\[\\mathbf{f}\] \\mathsf{M}\_{\\rho}\[\\mathbf{f},\\mathcal{L}^{-1}\\mathbf{f}\])\\\\  &=-2\\mathrm{tr}(\\mathsf{M}\_{\\rho}\[\\mathcal{L}\\mathbf{g},\\mathbf{g}\]) \+ \\mathrm{tr}(\\mathsf{M}\_{\\rho}\[\\mathcal{L}\\mathbf{g}\] \\mathsf{M}\_{\\rho}\[\\mathcal{L}\\mathbf{g},\\mathbf{g}\]).  \\end{aligned}  $$
>     The computational complexity remains the same. We note that, as we parameterize the $k$-th eigenfunction of $\\mathcal{L}^{-1}$ by $\\mathcal{L}g\_k$, at convergence, $\\mathcal{L}g\_k$ would correspond to the normalized eigenfunction $\\phi\_k$, and thus $g\_k(\\mathbf{x})=\\frac{1}{\\lambda\_k}\\phi\_k(\\mathbf{x})$ should ideally hold.
>
>     - This enables OMM to be used with a broad class of operators whose spectra diverge, including the Laplacian and other elliptic operators. We applied this technique to the 2D infinite well and 2D harmonic oscillator problems which have analytical solutions, and observed that OMM can successfully recover the top \~30 eigenfunctions accurately, to the similar order of what's reported for the 2D hydrogen atom. We will update the manuscript with these experiments. We will also add an ablation study for higher-dimensional versions of these basic problems to demonstrate the scalability.
>
> ---
>
> If these clarifications adequately address the reviewer’s concerns, we would greatly appreciate it if the reviewer could consider updating the score to reflect what we believe are meaningful contributions to the community. We would be happy to incorporate  any additional suggestions and comments, if any.

---

> > ### Comment · Reviewer_YTWU · 2025-08-05
> >
> > The authors have addressed my comments and questions. I will revise my score upward.

---

> > > ### Author Response · Authors · 2025-08-05
> > >
> > > We are pleased that our rebuttal has addressed your concerns. We would like to once again thank you for your support and constructive comments, which have been valuable in improving the manuscript.

---

### Official Review · Reviewer_9r7G · 2025-07-03

**Clarity:** 3
**Significance:** 3
**Originality:** 2
**Rating:** 3
**Confidence:** 2

**Summary:**

The paper includes three experiments:
1) Successor (Laplacian) representation in grid environments, which both ALLO and OMM can solve almost perfectly, unless when the spectral gap is extremely small.
2) Solve the 2D hydrogen atom Schrödinger equation, that a Sanger variant of OMM solves almost exactly, like NeuralSVD.
3) Contrastive representation learning (by learning eigenfunction of the canonical dependence kernel of random views) on CIFAR-100, achieving ~64% top-1 accuracy with a higher-order variant (OMM p=2), compared to SimCLR's ~66.5%

**Questions:**

- Can the authors provide experiments where  OMM has advantages over existing methods? Or argue for which properties a problem should have for OMM to be advantageous?
- What is the computational cost of OMM compared to other methods, especially with p=1 or higher-order variants?

Minor notes:
- There seems to be a small typo in line 62 in $\mathcal{R}^{d_1}$
- Add to the caption of figure 2: what is depicted (solution of the H2 Schroedinger equation) and what to take away from it (that both learn the same true representation?)

**Ethical Concerns:**

["NO or VERY MINOR ethics concerns only"]

**Final Justification:**

I raise my overall score to 3, and the significance rating to 3. I slightly recommend rejection as I do not find the advantages of OMM apparent from the theory or experiments, but remain low confidence as I am not very familiar with the matter, specifically the use of OMM in quantum chemistry.

**Limitations:**

yes

**Quality:**

3

**Strengths And Weaknesses:**

Strengths:
- The paper is clearly written, and could serve as a useful resource for anyone who wants to explore OMM
- The paper transparently discusses challenges in application and optimization (non-PSD with OMM are only possible if the eigenvalues can be bounded, numerical instability with near-zero eigenvalues, p=1 vs higher-order)


Weaknesses:
I understand the goal of the paper is to motivate, rather than beating baselines. Nevertheless, I currently I do not find it very motivating:
- The experiments do not show that OMM has any advantages over the baseline
    - The successor representation learning on GridMaze and GridRoom seems saturated. Both OMM and the baseline perform equally well, with a close to perfect cosine similarity of one, except for poor performance on few environments where the spectral gap is extremely small.
    - Similarly the 2D hydrogen atom seems to be perfectly solved by the baseline. Does OMM have an advantage for other Hamiltonians?

- From the discussion it is not clear to me when OMM would be useful. The only advantage seems to be that the baselines lack theoretical convergence guarantees?

---

> ### Author Rebuttal · Authors · 2025-07-31
>
> We appreciate the reviewer's time and effort in evaluating our manuscript. The primary concern raised appears to be that the advantages of OMM over existing methodologies are not sufficiently clear. Below, we clarify the theoretical and practical merits of OMM in comparison to existing methods, and explain why it warrants further attention from the community. We hope this will allow the reviewer to reassess the contribution of our work, and we will carefully update our manuscript accordingly.
>
> ---
>
> 1. **On Theoretical Merits**
>
> Before addressing the concerns related to experiments, we emphasize why repositioning OMM represents a significant contribution. Although numerical linear algebra is a well-established field, researchers in diverse areas (such as computational physics and chemistry for solving linear PDEs, statistics for correlation analysis, theoretical computer science for streaming PCA/CCA, and machine learning for representation learning) have often developed their own computational methods. These methods are sometimes tailored to specific types of matrices or operators relevant to their domains, and sometimes these methods can be more generic and broadly applicable in disguise. In many cases, however, such methods inadvertently rediscover techniques already developed in other fields, often in more primitive or suboptimal forms.
>
> As a particular example developed in the field of density functional theory, OMM has been substantially studied in specialized cases, with some highlights by numerical linear algebraists (see, e.g., \[9\], \[29\]). **However, the idea has never been explicitly considered as a generic tool for spectral decomposition of PSD operators**, which may result in efficient algorithms for various problems including streaming PCA, learning representation for RL, and solving PDEs, as we highlight in the manuscript. While we acknowledge that our experimental results do not outperform some existing baselines, the experiments show competitive, if not outperforming,  performance to the state of the art methods developed for such problems.
>
> Below, we highlight the theoretical and conceptual merits we bring by repositioning the OMM.
>
> - `Within streaming PCA literature`: As we alluded to in Section 2.2.1, the literature for streaming PCA has devoted subsequent effort in trying to develop an efficient algorithm that learns top-k “orthogonal” eigenvectors. To our understanding, however, the literature unfortunately has narrowly focused on Oja’s algorithm and its variants or constrained optimization framework such as Rayleigh quotient (RQ) maximization (in Eq. (1)), both of which require explicit handling of orthogonality constraints. This inherently complicates their analysis and limits applicability to more high-dimensional problems. One recent example is EigenGame \[11\], which won a best paper award from ICLR 2021; however, EigenGame still requires explicit normalization, derived from the RQ maximization framework.
>
>     Given this, we found it noteworthy that OMM can result in a simpler and more efficient (i.e., without explicitly handling orthogonality constraint) algorithm. Thanks to the implicit orthogonalization embedded in OMM, this can potentially lead to more scalable spectral algorithms for large-scale, high-dimensional problems. We note that LoRA has the same property and benefit (see Section 2.4), and its generic modern application was studied in \[41\].
>
> - `Within representation learning for RL literature`: In RL, there exists a rather independent literature for the “successor representation learning” framework. Their literature stems from the graph drawing algorithm (GDO), which essentially aims to solve the RQ maximization in Eq. (1), where the orthogonality constraint is handled via a Lagrangian multiplier. It was later generalized to the “generalized GDO (GGDO)” to learn top-k eigenvectors in the order of eigenvalues. ALLO \[13\] was proposed to improve GGDO, removing the need of tuning the Lagrangian multiplier in GDO/GGDO, by the augmented Lagrangian method. We emphasize, however, that (1) ALLO needs to solve max-min problem which require tuning two separate learning rate scales, and (2) ALLO still has a hyperparameter to be tuned, the barrier parameter $b$, which is mentioned in the paper that they proposed to increase it in a gradient ascent fashion as a heuristic. In contrast, OMM is a simple minimization framework, and does not have any parameters that must be tuned or handled heuristically. In practice, the parameter-free nature allows a much easier implementation, which implies a wider applicability of OMM than ALLO. In direct contrast, our experiment demonstrates that OMM can perform on par with ALLO, even with a much simpler idea and implementation.
>
>     - **Additional ablation study on ALLO**: To better highlight the issue of the hyperparameter tuning, we performed an additional ablation study for the RL experiment, by increasing d=11 to d=50. In summary, we demonstrate that the optimal hyperparameters of ALLO (the initial value of $b$ and the learning rate to update $b$) tuned for d=11 result in poor performance for d=50; this suggests that ALLO requires expensive hyperparameter tuning for not just for new problem instances, but even for different numbers of modes to be retrieved. In contrast, OMM was able to successfully train the network with the same optimization configuration for d=11. This provides a strong case for why OMM may be preferred over ALLO in general. We will include this new ablation study in our revision.
>
> 2. **Questions**
>
> - `Can the authors provide experiments where OMM has advantages over existing methods? Or argue for which properties a problem should have for OMM to be advantageous?`
>
>     As explained above, we demonstrate how OMM performs comparably to ALLO, without any additional technique such as the augmented Lagrangian. In the 2D hydrogen example, we demonstrate how OMM performs comparably to LoRA (also known as NeuralSVD, see [41]), which was shown in \[41\] to outperform existing methods such as spectral inference networks \[A\] and NeuralEF \[B\].
>
>     The reviewer might question then when OMM may stand out compared to LoRA. As alluded to in line 239, we include an experiment in Appendix C.3.2 to demonstrate when LoRA can miserably fail, while OMM can still learn good representation. In a nutshell, LoRA may not be able to learn eigenfunctions when the target matrix/operator to decompose is derived from a “sparse” kernel. We explain the reasoning behind this failure mode below.
>
>     - Consider an operator defined by a kernel. Recall the objectives of OMM and LoRA after line 227, and note that the term $\\mathrm{tr}(\\mathsf{M}\_{\\rho}\[\\mathbf{f},\\mathcal{T}\\mathbf{f}\])=\\mathbb{E}\_{\\rho(\\mathbf{x})\\rho(\\mathbf{x}')}\[\\mathbf{f}(\\mathbf{x})^\\intercal k(\\mathbf{x},\\mathbf{x'}) \\mathbf{f}(\\mathbf{x}')\]$ appears only in the first term of LoRA objective. Given this, suppose that the kernel is extremely sparse, i.e., $k(\\mathbf{x},\\mathbf{x}')\\approx 0$ with high probability for $\\mathbf{x},\\mathbf{x}’\\sim p(\\mathbf{x})p(\\mathbf{x}')$. (This can happen when the underlying matrix is extremely sparse, like a graph Laplacian matrix of a real-world graph.) In such cases, if we perform stochastic gradient updates with minibatch samples, the optimization signal from the LoRA objective will only be based on the second term, which only encourages orthogonality. By contrast, OMM can still learn useful signals in this scenario, as the term $\\mathrm{tr}(\\mathsf{M}\_{\\rho}[\\mathbf{f},\\mathcal{T}\\mathbf{f}])$ appears both terms in the OMM objective and thus the scales of the two terms are “balanced”.
>
>     - We demonstrate this case for a real-world graph Laplacian in Appendix C.3.2, and evaluate the quality of representations in the downstream task of node classification. In this example, we observe that LoRA failed to converge during training, while OMM demonstrates decent linear probe performance on the task. It even outperforms one baseline (Neural Eigenmaps), demonstrating the outstanding discriminative performance of learned eigenfunctions (marked as “embedding”), whereas the learned eigenfunctions in Neural Eigenmaps do not show good discriminative power and thus they had to resort to the intermediate layer representation (marked as “representation”).
>
>     We believe that this demonstrates the superiority of OMM against the existing parametric spectral solvers such as NeuralSVD (based on LoRA) and Neural Eigenmaps.
>
>     ```
>     [A] Pfau, David, et al. "Spectral inference networks: Unifying deep and spectral learning." ICLR, 2019.
>     [B] Deng, Zhijie, Jiaxin Shi, and Jun Zhu. "NeuralEF: Deconstructing kernels by deep neural networks." ICML, 2022.
>     ```
> - `What is the computational cost of OMM compared to other methods, especially with p=1 or higher-order variants?`
>
>     In terms of complexity in computing a given objective, OMM’s complexity is similar to ALLO and LoRA when $p=1$, where the dominant factor is in computing the empirical moment matrices, which takes $O(BK^2)$ complexity for a given minibatch of size $B$ and for $K$ eigenfunctions. When $p\>1$, the complexity becomes $O(BK^2+pK^3)$, as we need to explicitly compute the matrix powers in Eq. (7).
>
> 3. **On Minor Notes**
>
> We appreciate the detailed suggestions after careful reading. We will revise the paper accompanying the comments.
>
> ---
>
> If the responses provided address the reviewer's concerns, we would appreciate reconsideration of the score, as we believe the paper makes significant contributions to the field.

---

> > ### Author Response · Authors · 2025-08-05
> >
> > Dear Reviewer 9r7G,
> >
> > Thank you again for your effort in evaluating our paper. As the discussion period is getting to a close, we would like to kindly follow up and ask whether our rebuttal has sufficiently addressed your concerns, or if there are any remaining points you would like us to clarify.
> >
> > We would be happy to further engage with you if needed.
> >
> > Sincerely,\
> > The Authors of Submission 4948

---

> ### Comment · Reviewer_9r7G · 2025-08-08
>
> I thank the authors for their response. They partially addressed my concerns of the advantages of OMM over established methods. As they pointed out, the advantage of OMM over ALLO in representation learning is that OMM does not require an extra hyperparameter and optimization is simpler. The authors also show a failure case of LoRA where OMM outperforms in appendix C.3.2
>
> Ultimately, I still find the experiments in the main text are not very convincing, since the tasks seem to be so easy, that the baselines methods can already solve them almost perfectly. I would recommend the authors to add harder tasks and highlight the failure case of LoRA in appendix C.3.2. I raise my overall score to 3.

---

### Decision · Program_Chairs · 2025-09-17

**Decision:**

Accept (poster)

**Comment:**

Machine learning has a long and fruitful history of translating methods from other field to the context of machine learning, and this paper does so for a spectral method known as orbital minimization in computational chemistry.

The paper is well-written and makes an interesting contribution in a topical area. The reviews and the discussion clearly support publication. The revisions promised by the authors, in particular the promised ablation studies on spectral shifts, hyperparameter sensitivity etc, will strenghten the paper further.